# Salience memories formed by value, novelty and aversiveness jointly shape object responses in the prefrontal cortex and basal ganglia

**Ali Ghazizadeh** [1] ✉ **& Okihide Hikosaka** [2]

Ecological fitness depends on maintaining object histories to guide future interactions. Recent evidence shows that value memory changes passive visual responses to objects in ventrolateral prefrontal cortex (vlPFC) and substantia nigra reticulata (SNr). However, it is not known whether this effect is limited to reward history and if not how cross-domain representations are organized within the same or different neural populations in this corticobasal circuitry. To address this issue, visual responses of the same neurons across appetitive, aversive and novelty domains were recorded in vlPFC and SNr. Results showed that changes in visual responses across domains happened in the same rather than separate populations and were related to salience rather than valence of objects. Furthermore, while SNr preferentially encoded outcome related salience memory, vlPFC encoded salience memory across all domains in a correlated fashion, consistent with its role as an information hub to guide behavior.

Our past experience with objects or lack thereof is a critical determinant of our interactions with our environment. Objects which predicted rewards or punishments in the past often invoke strong motivational and emotional responses in humans and animals alike[1–4]. Novel and never-seen-before objects often stimulate curiosity and encourage explorations[5–7]. Notably, all such interactions rest on the ability of the animal to form appropriate memories for each object and to recall those memories in future interactions with that object. There are at least two alternatives for the way that object's past history can affect neural processes that guide behavior. One possibility is that object memories reside in some brain regions (memory circuit) and are loaded and accessed by circuits controlling behavior (action circuit) only when needed in a goal-directed fashion. Another possibility is that past histories of objects change their very visual responses within the action circuit even during passive exposures and without active interactions with the object.

One such action circuitry is the loop between prefrontal cortex and basal ganglia which is involved in gaze control[8,9]. Prefrontal cortex is known to be sensitive to object novelty[10,11] even from childhood[12]. Prefrontal cortex is also implicated in processing rewarding and aversive stimuli[13–15]. Likewise selective coding of rewarding and/or punishing stimuli is observed in basal ganglia[3,16,17]. In addition, novel objects are also known to invoke enhanced responses in areas within basal ganglia such as in caudate[18]. We have previously shown that value memories shape the visual responses of objects similarly and with a remarkable granularity in vlPFC and the basal ganglia output, SNr.[19]. However, it is not known whether the visual responses to objects in the corticobasal circuitry are also affected by other past experiential dimensions such as perceptual exposure (familiarity vs novelty) or aversive associations and if so whether the same neurons encode such disparate object memories across different domains.

It is often assumed that reward, novelty or punishment enhance an object's importance and relevance for an animal both motivationally and emotionally (aka object's salience). However, in many cases such physiological reactions are presumed without direct measurements and the exact meaning of salience is not fleshed out[16,20,21]. Here,

[1]Bio-intelligence Research Unit, Electrical Engineering Department, Sharif University of Technology, Tehran 11365-11155, Iran. [2]Laboratory of Sensorimotor Research, National Eye Institute, NIH, Bethesda, MD 20892, USA. ✉e-mail: alieghazizadeh@gmail.com

we consider a precise working definition of salience as the attentional bias toward an object which is quantified by measuring gaze bias during free viewing of multiple competing objects[5]. Such attentional bias conceptualization of salience has been used previously[3,22,23]. In this case, if novelty and aversiveness are found to affect the gaze bias behavior similar to value one may conclude them to have a positive salience. Furthermore, since this attentional bias happens due to past experience and seen in the absence of rewarding or punishing expectations in free viewing, we refer to it as salience memory of an object. In such a case a neuron is encoding salience if it responds similarly to objects with similar salience regardless of their valence.

To address these questions, we recorded the single unit activity across multiple domains (appetitive, aversive, novelty) within the same neurons in vlPFC and SNr. This allowed us not only to address whether domains other than value are represented in vlPFC and SNr but also to reveal joint-coding of object memories across domains in the corticobasal circuitry and the regional similarities and differences. In brief, neurons in vlPFC were found to be sensitive to all three domains examined while SNr neurons were more tuned to objects with appetitive and aversive outcomes and less concerned with familiarity/novelty domain. Notably, neural activations paralleled the measured salience of objects during free viewing rather than the valence of objects measured in binary choice thus providing strong evidence for salience memory coding in the vlPFC and SNr.

## Results

To examine and contrast the role of prefrontal cortex and basal ganglia across domains of past experience with objects, acute neural recordings from ventrolateral PFC (vlPFC, areas 8Av, 46 v, and 45) and from caudo-dorsolateral substantia nigra reticulata (cdlSNr) were done in separate sessions in two macaque monkeys (monkeys B and R). Both subjects were previously trained with abstract fractal objects that had appetitive or aversive outcomes or just became familiar without any outcome association across multiple sessions (>5 sessions and >8 repetitions per object per session prior to start of recording for each fractal type, Methods)[5]. To test whether the memory of these past experiences with objects is reflected in the visual responses of the corticobasal circuitry, neural responses to objects were recorded using a passive viewing procedure in the absence of any outcome for objects and in the absence of saccades (Methods)[19].

We have previously shown that apart from the opposite polarity and differences in value signal magnitude, vlPFC and cdlSNr show remarkable similarities in encoding memory of object-reward associations (value memory) with a high-level of granularity[19]. Both vlPFC and cdlSNr have projections to superior colliculus (SC) and thus can work synergistically to promote finding and orienting toward valuable objects. Given the role of prefrontal cortex in novelty processing[24] and the fact that novel objects also attract attention[5], we wondered whether the similarity of vlPFC and cdlSNr responses extends to novel vs familiar object contrasts (perceptual memory).

To compare the effect of value memory and perceptual memory, visual responses were recorded with high vs low value (good vs bad) objects and novel vs familiar objects within the same neurons in each region (Methods). Good and bad objects were selected from sets that were previously trained with high or low reward amount (sets of 8 objects: 4 good/4 bad, Fig. 1a). Familiar objects were chosen from among fractals that became perceptually familiar to the monkeys over multiple sessions of passive viewing but with no reward pairing (>5 sessions). Novel objects for each neuron were picked from never-seen-before fractals (sets of 8 objects: 4 novel/4 familiar, Fig. 1a). A passive viewing block was done with either good/bad objects or novel/familiar objects. During the passive viewing task, objects were shown pseudo-randomly in the neuron's receptive field (Fig. 1b).

Figure 1c, d show average population responses of vlPFC and cdlSNr neurons to good/bad and novel/familiar objects (average

PSTHs). As reported previously vlPFC neurons showed on average stronger excitation to good objects (for average AUC see Fig. 1g, h). SNr showed robust excitation to bad objects and an equally robust inhibition to good objects with drastic differential response to good/bad objects (significantly stronger than vlPFC, $t_{233} = 12$, $p < 1e-3$). Interestingly, responses of the same neurons recorded with novel/familiar object revealed a somewhat different pattern. While vlPFC neurons on average showed stronger excitation to novel compared to familiar objects, paralleling responses to good and bad objects, for cdlSNr the response difference to novel vs familiar objects was muted when compared to differential activations seen to good and bad objects (see Supplementary Fig. 1, for example rasters). The difference in sensitivity to value vs novelty can be further examined by comparing value (good minus bad response) and novelty (novel minus familiar response) signals within each region (Fig. 1e, f). In vlPFC the novelty signal peak was about 55% of value signal peak ($5.6 \pm 0.8$ Hz vs $10.1 \pm 1.5$ Hz). In cdlSNr the novelty signal peak was only about 13% of value signal peak ($7.7 \pm 1.7$ Hz vs $56 \pm 4.4$ Hz). Furthermore, while the distribution of onset times of value and novelty signal were comparable in vlPFC, in cdlSNr, the onset of novelty signal was significantly later than the value signal (Fig. 1e, f). These value and novelty signals and their onsets in vlPFC and cdlSNr were consistent in both monkeys (Supplementary Fig. 2).

While the average PSTH shows value and novelty signal within the same population, it does not address how the signal strength vary across value and novelty domains for individual neurons. To measure joint-coding of novelty and value memories, the AUC for discrimination of novel and familiar objects (novelty AUC) for each neuron is plotted against the AUC for discrimination of good and bad objects (value AUC) (Fig. 1g, h). Consistent with average PSTHs (Fig. 1c, d), in vlPFC most neurons showed significantly stronger excitation to good objects (49% good-preferring neurons or Gp) and novel objects (40% novel-preferring neurons or Np) compared to the percentage with stronger excitation to bad objects (14% bad-preferring or Bp) or to familiar objects (8% familiar-preferring or Fp). Importantly, in addition to the significant bias toward good and novel preference in the marginal distributions, there was a significant correlation between novelty and value AUCs across the vlPFC neuronal population (Fig. 1g). The joint distribution of cdlSNr portrayed a different picture. The value AUC indicated a makeup of predominantly Bp neurons (70% Bp, 4% Gp) while novelty AUC showed a more balanced representation of Np and Fp neurons (32% Fp, 20% Np). The marginal distribution of value was strongly and significantly skewed toward bad-preference but the marginal distribution of novelty was only slightly skewed toward familiarity-preference. Additionally, and unlike vlPFC, the correlation between novelty and value coding across cdlSNr neurons was not significant (Fig. 1h). Interestingly, the strong value coding and the lack of novelty coding or value-novelty correlation were also observed in the neighboring putative dopaminergic neurons recorded in the substantia nigra compacta (SNc) (Supplementary Fig. 3).

One interpretation about the weaker novelty signal in cdlSNr might be that it cares more about past experiences that create object outcome associations rather than mere perceptual exposures. If so, aversive associations should be well represented in cdlSNr similar to what is observed for appetitive associations with reward. Using objects with past aversive outcomes is also important for addressing whether the corticobasal circuitry is exclusively concerned with value memory or more generally with salient objects regardless of valence. To address these issues, additional objects were trained using a Pavlovian procedure either with reward (good objects), airpuff (aversive objects) or no-outcome (neutral objects) (Fig. 2a) and then tested using the same passive viewing procedure to check for lasting effects of past training on their visual responses. Following training, choice trials were used to check for object's relative valences among the three categories

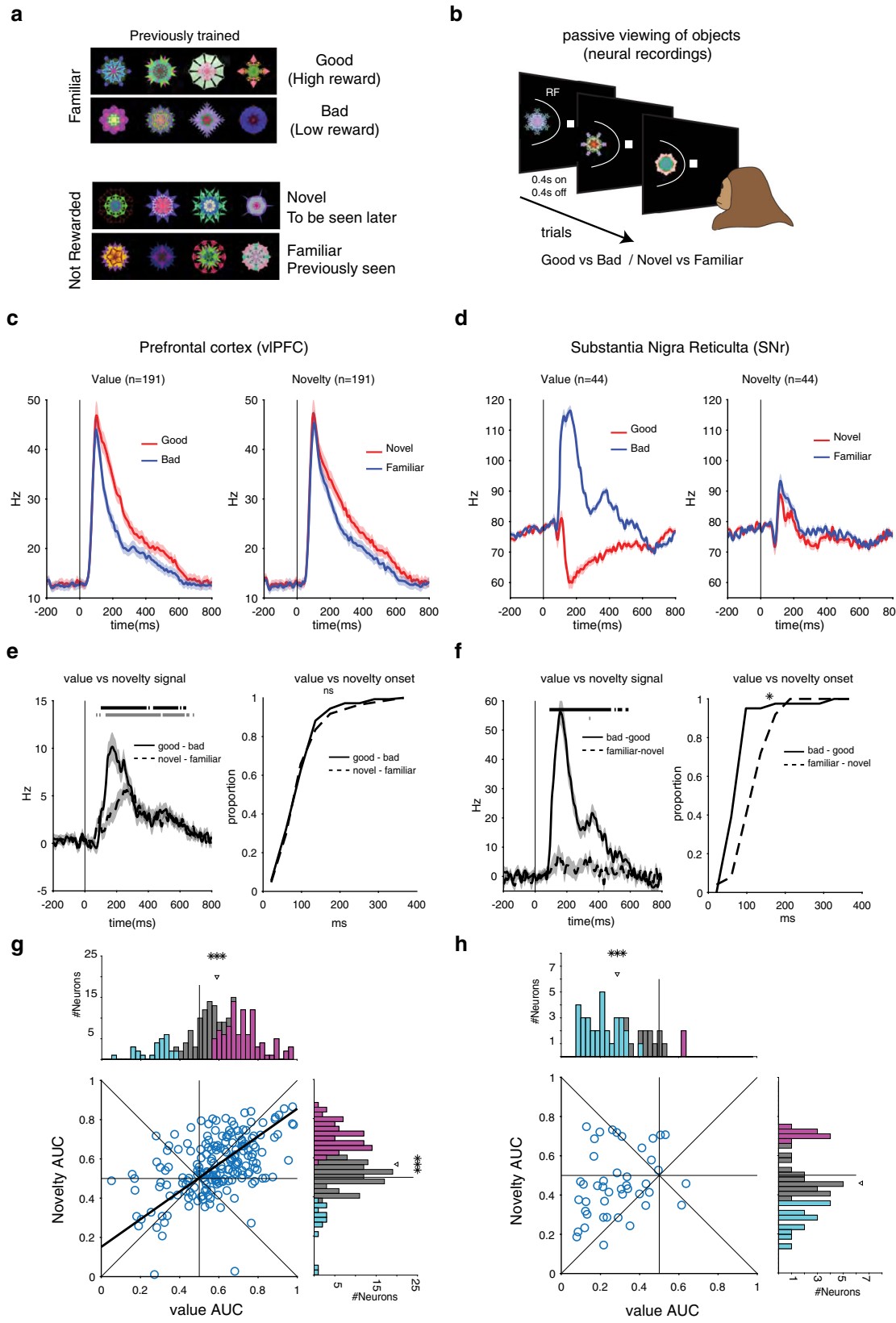

(good, neutral and aversive, Fig. 2b). Despite their negative valence, aversive objects are often presumed to be attentionally salient. Indeed, many studies consider opposite activations to rewarding vs aversive objects in a region as an evidence for valence coding and coactivation with the same polarity as evidence for salience coding[3,20,25]. Importantly, in such cases aversive salience is often presumed 'without'

formal behavioral testing. To do away with such presumptions and to gauge the degree of learned salience, a free viewing procedure was used with objects randomly drawn from good, aversive or neutral categories (Fig. 2c, Methods). The saccade bias toward objects upon display onset was used to quantify the relative salience among the object categories[5].

**Fig. 1 | Novelty and value signals are significant and correlated in vlPFC but are uncorrelated in cdlSNr despite strong and significant value coding. a** The value sets consisted of objects with history of small and large rewards (bad and good objects, respectively). Good and bad objects were equally familiar. The perceptual sets consisted of objects with history of passive visual exposure and never-before-seen objects (familiar and novel objects, respectively). Familiar and novel objects had no reward history. **b** Passive viewing task: monkeys kept central fixation while objects from a given set were randomly and sequentially shown within the neuron's receptive field (RF) in the absence of saccades and object outcomes. **c** Population average peristimulus time histogram (PSTH) to good vs bad objects and to novel vs familiar objects in vlPFC. Color-patch indicates s.e.m. here and thereafter. **d** Same format as C but in cdlSNr. **e** Average firing difference between good vs bad objects (value signal: good minus bad) and between novel vs familiar objects (novelty signal: novel minus familiar). The black and gray horizontal lines indicate times of significant difference from zero ($p < 0.01$) for value and novelty signals, respectively

(left plot). Cumulative distribution of value signal and novelty signal onsets across neurons in vlPFC ($t_{206} = -0.5$, $p = 0.5$) (right plot). **f** Same format as E but in cdlSNr. The value and novelty signal are inverted in cdlSNr (left plot). For the cumulative distribution of value signal and novelty signal onsets ($t_{52} = -2.3$, $p = 0.02$) (right plot). **g** Marginal and joint distribution of novelty vs value AUC across all neurons in vlPFC (Pearson's $\rho = 0.5$ $P = 1e{-}13$, value AUC $= 0.59$, $t_{190} = 7$, $P = 4e{-}11$, novelty AUC $= 0.56$, $t_{190} = 6.2$, $P = 2e{-}9$). **h** Same format as G but in cdlSNr (Pearson's $\rho = 0.27$, $P = 0.07$, value AUC $= 0.28$ $t_{43} = -9.5$, $P = 3e{-}12$, novelty AUC $= 0.46$ $t_{43} = -1.6$, $P = 0.11$). Regression line is plotted when correlation is significant (Deming regression). Arrows on marginal AUCs mark population average. AUC $> 0.5$ indicates higher firing to good/novel objects. Neurons with significant AUC $> 0.5$ and $<0.5$ and non-significant neurons are color coded in the marginal histogram. ns: not significant, $*p < 0.05$, $**p < 0.01$, $***p < 0.001$. All tests are two-sided here and thereafter. Source data are provided as a Source Data file.

Choice performance, following training (>5 sessions per set), showed robust preference for good over aversive (airpuff) or neutral objects as expected. In choices between neutral and airpuff objects there was also a significant preference for neutral object in both monkeys (Fig. 2d, e). This means that in both monkeys airpuff objects had a 'lower' valence compared to neutral objects (Fig. 2d, e). The choice bias was stronger in Monkey B compared to Monkey R (Monkey B %98, Monkey R %68). Nevertheless, and despite their lower valence, free viewing showed a significantly larger gaze bias toward airpuff compared to neutral objects in monkey B. In monkey R there was also a positive trend toward higher salience for airpuff vs neutral objects that did not reach significance. These results suggest airpuff objects to have an overall positive salience (higher gaze bias than neutral) despite their negative valence (lower value than neutral). Analysis of saccade reaction time for airpuff, neutral and good objects also showed faster saccades toward airpuff objects compared to neutral objects similar to saccade reaction time to good objects consistent with attentional salience observed in free viewing (Supplementary Fig. 4).

These observations make interesting predictions about what to expect from the visual responses in vlPFC and cdlSNr. If the visual responses are primarily concerned with valence, then in vlPFC the average excitatory response to the airpuff objects should be 'weaker' than the neutral objects. In cdlSNr given the lower valence of airpuff compared to neutral objects one should observe a 'stronger' excitation compared to neutral objects. If on the other hand, the visual responses in either region is shaped by learned salience then responses to airpuff should resemble visual responses to good objects at least in monkey B. Figure 2f, g shows results that are consistent with the latter possibility. For monkey B who attributed higher than neutral salience to airpuff objects, in vlPFC excitation to airpuff was much larger than neutral objects and close to good objects and in cdlSNr response to airpuff showed a strong inhibition similar to good objects. In monkey R, in vlPFC the excitatory response to airpuff was still slightly stronger than to neutral objects and in cdlSNr, the airpuff response was excitatory but below the neutral objects, consistent with the slightly higher salience of airpuff compared to neutral in this monkey. The differential neural response to airpuff vs neutral (airpuff minus neutral) tends to be positive in vlPFC and negative in cdlSNr in both monkeys but stronger in monkey B consistent with the higher salience of airpuff objects in this monkey. Thus, in neither monkey, we observed responses that were consistent with the negative valence of the airpuff objects. To be consistent with value since both monkeys showed negative value for airpuff compared to neutral (value of neutral > airpuff from binary choice, Fig. 2d, e), we should have seen higher firing to neutral objects compared to airpuff objects in vlPFC and lower firing to neutral objects compared to airpuff objects in cdlSNr but the opposite is observed (Fig. 2f, g). Note that our arguments about valence only rests on deciphering the 'ordinal' value of objects based on binary choices (i.e. if object A is chosen more than object B, object A has more

subjective value compared to B) and does not require knowledge of magnitude of their actual utility or subject value differences which cannot be estimated from binary choices without additional assumptions or experiments. We also note that the weaker choice and salience results for monkey R might be due to generalization of fearful response to neutral objects as evidenced by blinking during free viewing (Supplementary Fig. 6).

The full picture of airpuff vs reward memory coding in each region can be better examined once again by looking at the marginal and joint distributions across the populations (Fig. 3). In monkey B, the aversive signal (AUC of airpuff vs neutral) was significantly skewed toward airpuff preference in vlPFC and toward neutral preference cdlSNr consistent average PSTHs in this monkey. In monkey R, there was a similar but weaker trend toward airpuff preference in vlPFC and toward neutral preference in cdlSNr consistent with weak positive salience of airpuff in this monkey. Importantly, in monkey B for whom airpuff objects had positive salience, the aversive signal and value signal were positively correlated in both vlPFC and cdlSNr across the population. Together these results show that not only aversive object memory is encoded in the corticobasal circuitry, but that it is encoded by the same neurons which encode object value memory. Furthermore, the strength of value and aversive memories were positively correlated in both vlPFC and cdSNr across neurons when aversive objects have positive salience similar to good objects suggesting that neurons in both regions are signaling a common currency for salience regardless of the type of past outcome.

We have previously shown that not all aversive outcomes create positive salience compared to neutral objects[5]. If the salience theory of corticobasal activation were to hold, the response to aversive objects in aversive cases that are not salient should not be different from neutral objects responses. To test this hypothesis, object punishment history was created using aversive outcomes (aversive taste or time-out) that did not change object salience (Fig. 4a). Aversive taste and time-out were avoided in choice trials when paired against neutral objects (Fig. 4b, 88% saline, 78% timeout) showing their negative valence which was consistent separately for each monkey (Supplementary Fig. 5a, b). Unlike the airpuff objects, saline and time-out objects had similar salience to neutral objects or even a small trend toward negative salience (less salience than neutral) for the time-out objects using free viewing (Fig. 4c) and consistently for both monkeys (Supplementary Fig. 6). Importantly, as predicted by the salience hypothesis, responses to saline and time-out objects were similar to neutral objects (Fig. 4d). The aversive signal (aversive−neutral) showed a small negativity in vlPFC and a small positivity in cdlSNr (Fig. 4e). These neural responses for saline and time-out objects were consistent in both monkeys unlike the difference seen for airpuff objects (Supplementary Fig. 5c, d, for example units recorded with all three aversive outcomes see Supplementary Fig. 7). Consistently, the saccade reaction times to saline and time-out object was not different from

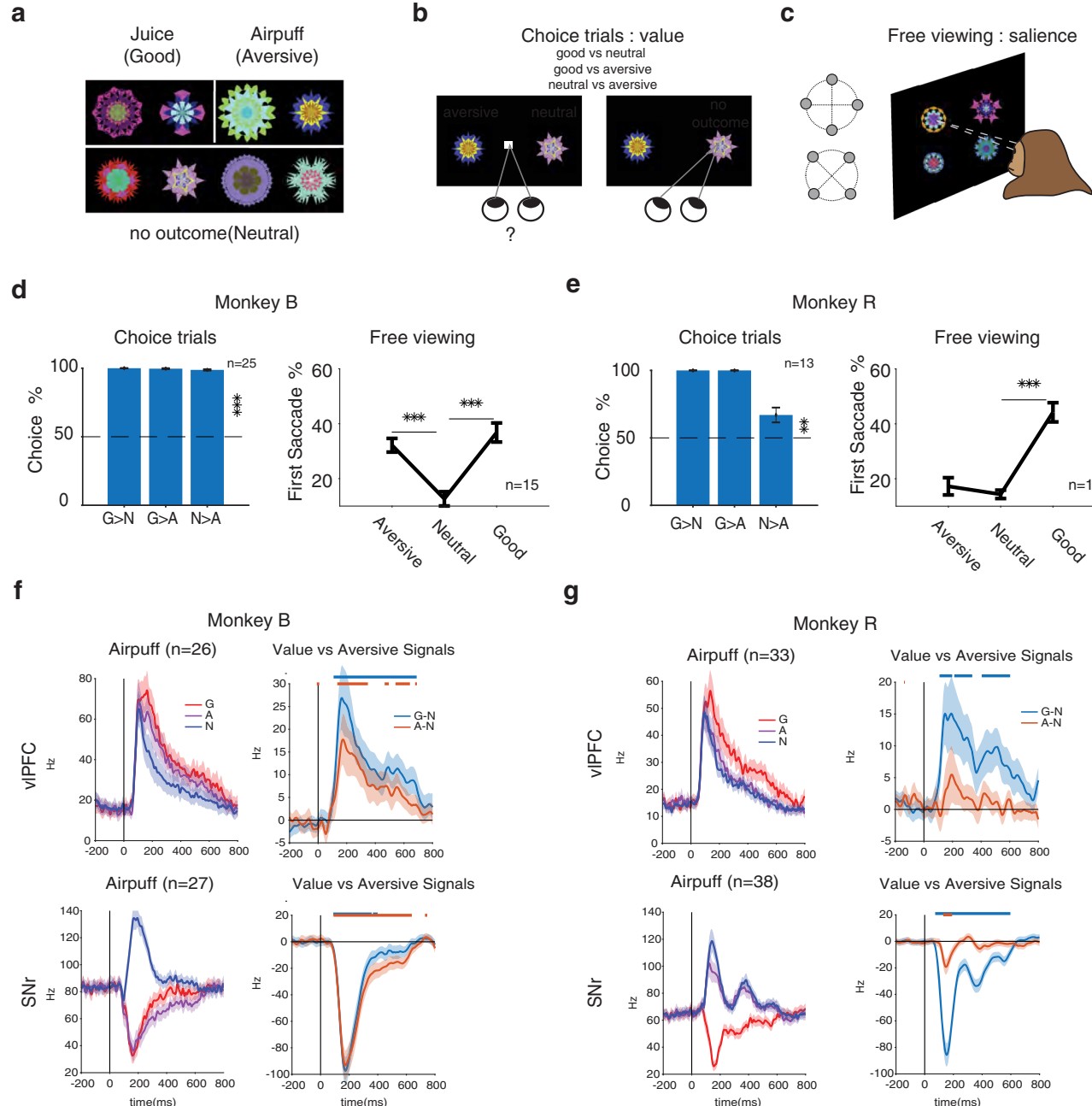

**Fig. 2 | Responses of vlPFC and cdlSNr to airpuff associated objects code their positive salience rather than their negative valence. a** The airpuff sets consisted of 2 objects with large reward (good objects) and 2 objects with airpuff (aversive objects) associations and 4 objects with no outcome (neutral). **b** Measuring valence: choice trials in which monkey denoted preference by saccading to and holding gaze on one object among two and received the associated outcome or no-outcome depending on the object category. Multiple saccades were allowed. **c** Measuring salience: free viewing trials in which 4 objects were randomly chosen from good, aversive and neutral categories and were displayed for free gaze with no outcome. **d, e** Average choice and free viewing performance for each monkey across sessions. Choice trial performance are separately shown for choosing good more than neutral (G > N), good more than aversive (G > A) and neutral more than

aversive (N > A) as a measure of valence (Monkey B: N > A, $t_{24}$ = 99 $p$ = 6e-33 Monkey R: N > A, $t_{12}$ = 3 $p$ = 9e-3, two-sided). Free viewing gaze bias using first saccade following display onset toward aversive, neutral and good objects as a measure of salience (Monkey B: $F_{2,42}$ = 19 p = 9e-7, aversive vs neutral $p$ = 1e-4, Monkey R: $F_{2,30}$ = 33 $p$ = 2e-8 aversive vs neutral $p$ = 0.7). **f, g** Population average PSTH to good, aversive and neutral objects and average value signal (good minus neutral) and aversive signal (aversive minus neutral) in vlPFC (top row) and in cdlSNr (bottom row) in passive viewing separately for each monkey. The horizontal lines indicate times of significant difference from zero ($p$ < 0.01) for value and aversive signals (right plots). Data are presented as mean values ± SEM. Source data are provided as a Source Data file.

neutral object and showed even a trend to be slower than neutral objects (Supplementary Fig. 4). The size of aversive signals in both regions were slightly larger for time-out objects consistent with its slight negative salience in free viewing. The marginal distribution of aversive AUCs was skewed toward neutral preference in vlPFC and skewed toward saline/time-out preference in cdlSNr. Nevertheless, the

joint distribution of value and aversive signals in this case did not show a significant correlation in either region (Fig. 5).

The salience account is also consistent with the observation that responses to novel/familiar objects were similar to good/bad responses given the higher salience of novel compared to familiar objects in both monkeys in free viewing (Supplementary Fig. 8). In addition, the

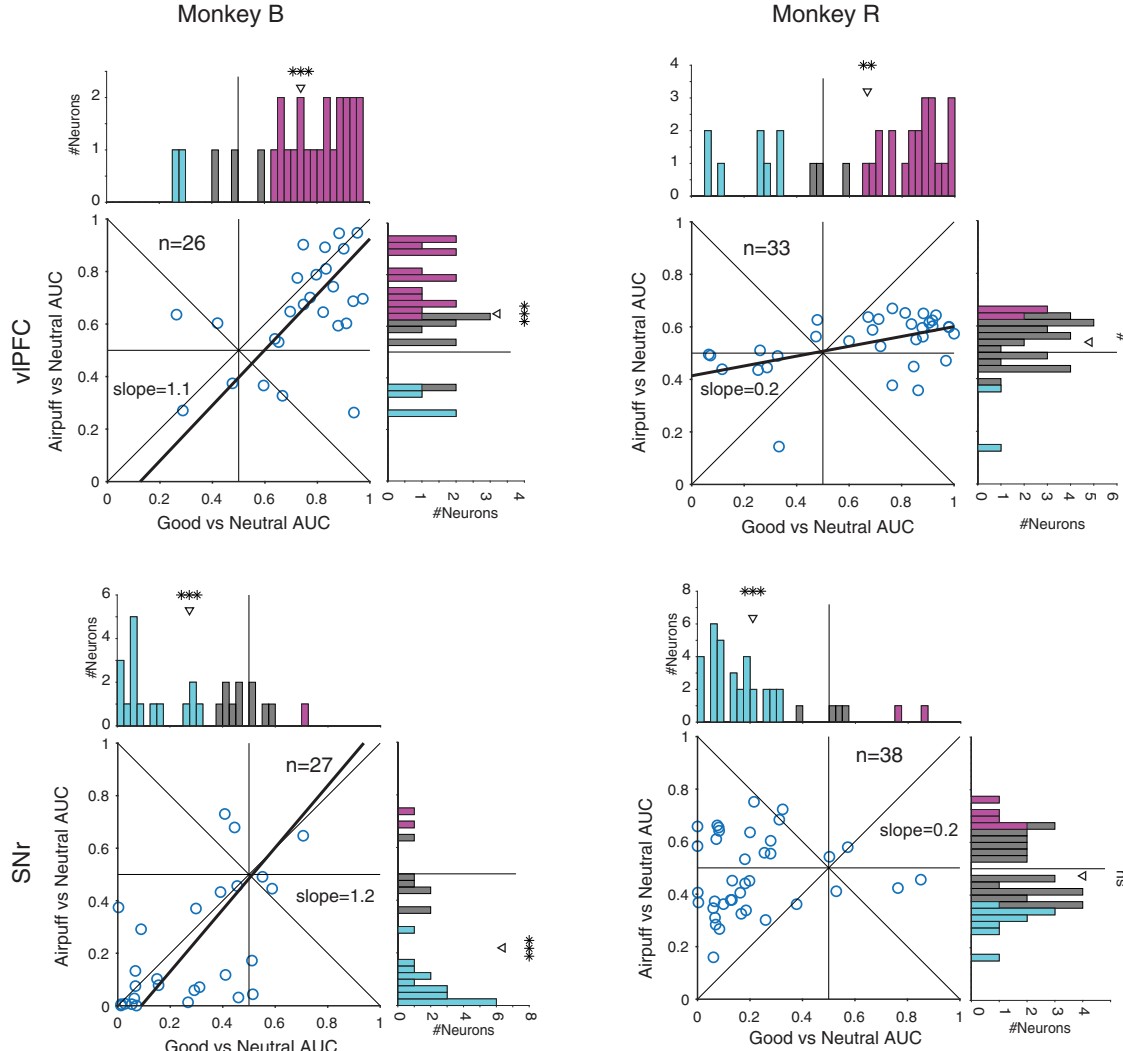

**Fig. 3 | Positive correlation between aversive and value signals in vlPFC and cdlSNr for airpuff associated objects which had positive salience.** Marginal and joist distribution of aversive AUC (>0.5: airpuff > neutral) vs value (>0.5: good > neutral) AUC across all neurons in vlPFC (top row) and in cdlSNr (bottom) separately for each monkey. (Monkey B, vlPFC Pearson's $\rho = 0.48$, $P = 1e-2$, value AUC = 0.73, $t_{25} = 6.1$, $P = 1e-6$, aversive AUC = 0.65, $t_{25} = 3.7$, $P = 9e-4$, cdlSNr:

Pearson's $\rho = 0.59$, $P = 1e-3$, value AUC = 0.27 $t_{26} = -5.5$, P = 8e-6, aversive AUC = 0.21 $t_{26} = -6.2$, P = 1e-6, Monkey R: vlPFC Pearson's $\rho = 0.45$, $P = 9e-3$, value AUC = 0.66, $t_{32} = 3.2$, $P = 2e-3$, aversive AUC = 0.53, $t_{32} = 2$, $P = 5e-2$, cdlSNr: Pearson's $\rho = 0.11$, $P = 0.5$, value AUC = 0.21 $t_{37} = -8.8$, P = 1e-10, aversive AUC = 0.47 $t_{37} = -1.14$, P = 0.2). Format is similar to Fig. 1g, h. #$p = 0.05$. Source data are provided as a Source Data file.

salience theory is consistent with neural responses in vlPFC and cdlSNr to objects with graded reward amount and probability reported previously[19]. In both cases, neural firings paralleled objects learned salience including the enhanced attention to uncertain rewards and lower attention to amount vs probability objects at the lowest and highest value extremes despite matching value and lack of uncertainty as measured by free viewing (Supplementary Fig. 9).

Results showed that object discriminability was mostly similar between main object categories across neuron types (good vs bad, familiar vs novel and aversive vs good objects) in both vlPFC and SNr (Supplementary Fig. 10, some exceptions in SNr: in aversive sets trending lower discriminability of neutral objects in Bp neurons and higher discriminability of good objects in NS neurons. In vlPFC: in good/bad and novel/familiar sets somewhat higher discriminability of good and novel objects in Gp neurons). This suggests that previous experience in appetitive, aversive or perceptual domains had modest if any effects on changing object selectivity within an object category in vlPFC and SNr which in any event were previously found to be low by measures such as sparsity[15] or nonuniformity[19].

To examine the temporal dynamics of correlated coding of cross-domain object salience, sliding pairwise correlation analyses time-locked to object onset across neurons between every two dimensions from among value, aversiveness and novelty were performed. Figure 6 shows that in vlPFC value signal (good–bad), novelty signal (novel–familiar) and aversive signal (airpuff -neutral) were all significantly correlated across the population. On the other hand, in cdlSNr, only the correlation between value signal and aversive signal was significant. Novelty signal was not correlated with either value or aversive dimensions. The correlation between value and aversive signal was much stronger in monkey B who actually perceived the aversive (airpuff) objects to be more salient (Supplementary Fig. 11). These results show that vlPFC neurons encode cross-domain object salience in a correlated fashion and thus signaling a common currency for object salience that includes value, aversiveness, novelty and even recency[15,19]. On the other hand, cdlSNr seem to be more concerned with outcome-related object salience but not novelty or recency (Supplementary Fig. 12, Table 1). Furthermore, results showed that in SNr the value signal seemed to have a faster onset compared to other domains such as novelty or aversive signals (Fig. 1f, Supplementary

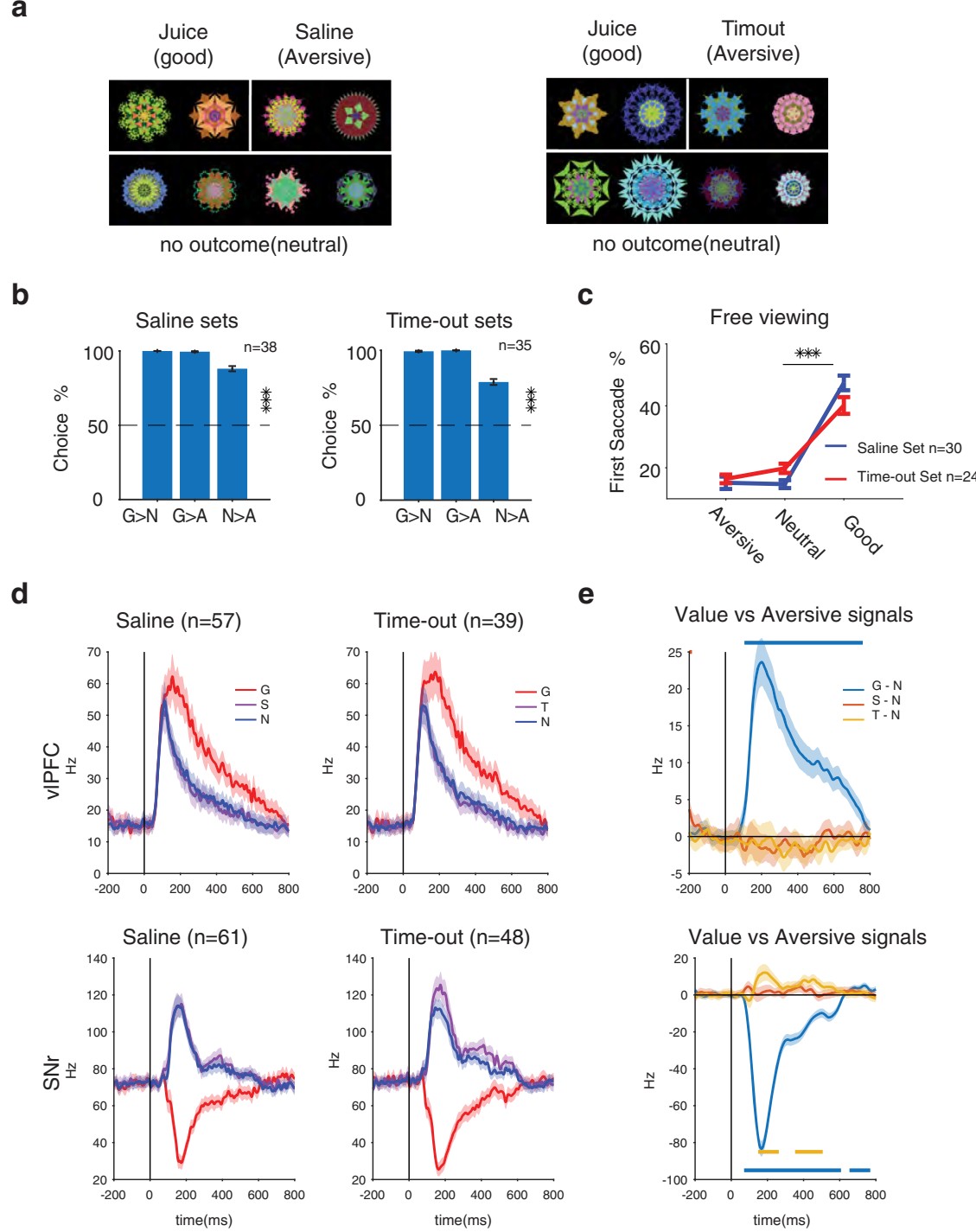

**Fig. 4 | Responses of vlPFC and cdlSNr to saline and timeout associated objects are consistent with their negligible salience rather than their negative valence. a** Same set structure as airpuff sets except for aversive objects. Saline sets consisted of 2 objects with saline outcome (left) and timeout sets consisted of 2 objects which delaying start of next trial (right) (**b**) Average choice between object pairs from three object categories (aversive, neutral and good) for saline and timeout sets across sessions and monkeys (saline: N > A, $t_{37} = 21$ $p = 1e{-}22$, $n = 38$ sessions, time-out: N > A, $t_{34} = 14$ $p = 5e{-}16$, $n = 35$ sessions). **c** Average free viewing gaze bias using first saccade following display onset toward aversive, neutral and good objects as a measure of salience (saline: $F_{2,86} = 92$ $p = 3e{-}22$, aversive vs neutral $p = 0.9$,

$n = 30$ sessions, timeout: $F_{2,68} = 42$ $p = 1e{-}12$, aversive vs neutral $p = 0.4$, $n = 24$ sessions). **d** Population average PSTH to good, aversive and neutral objects in vlPFC (top row) and in cdlSNr (bottom row) across monkeys for saline (left) and timeout (right) sets. **e** Average value signal (good minus neutral) and aversive signal (saline minus neutral and timeout minus neutral) in vlPFC (top row) and in cdlSNr (bottom row) across monkeys. The horizontal lines indicate times of significant difference from zero ($p < 0.01$) for value and aversive signals. Data in D and E are presented as mean values ± SEM. Statistical tests are two sided. Source data are provided as a Source Data file.

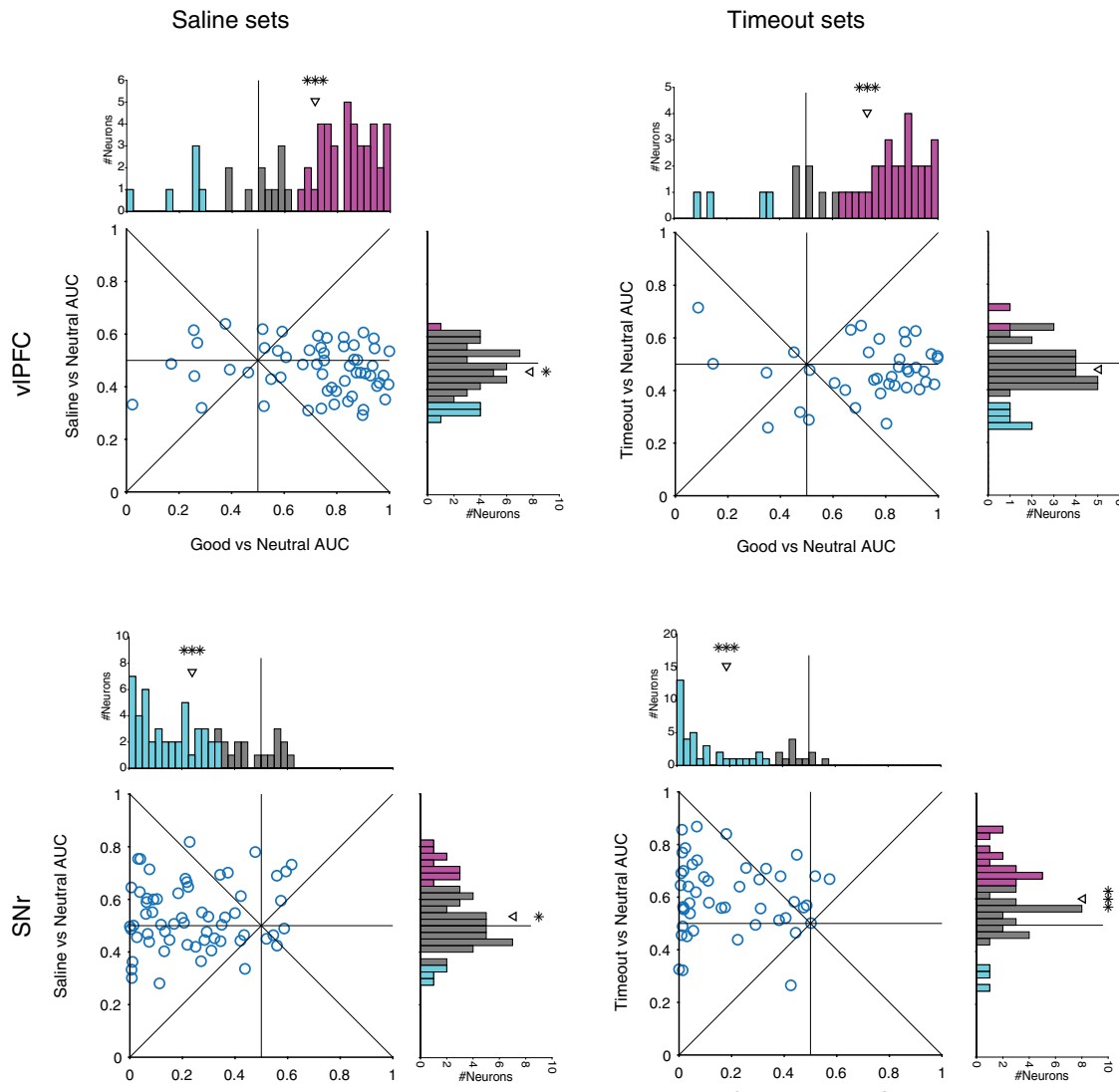

**Fig. 5 | Lack of correlation between aversive and value signals in vlPFC and cdlSNr for saline and timeout associated objects which had weak salience.** Marginal and joist distribution of aversive AUC (>0.5, left: saline > neutral, right: timeout > neutral) vs value (>0.5, good > neutral) AUC across all neurons in vlPFC (top row) and in cdlSNr (bottom). (Saline set, vlPFC Pearson's $\rho = -0.09$, $P = 0.49$, value AUC = 0.72, $t_{56} = 6.9$, $P = 4e-9$, aversive AUC = 0.47, $t_{56} = -2.6$, $P = 1e-2$, cdlSNr:

Pearson's $\rho = 0.12$, $P = 0.36$, value AUC = 0.24 $t_{60} = -11$, $P = 5e-16$, aversive AUC = 0.53 $t_{60} = 2.1$, $P = 3e-2$, Timeout set: vlPFC Pearson's $\rho = 0.03$, $P = 0.9$, value AUC = 0.73, $t_{38} = 6.3$, $P = 2e-7$, aversive AUC = 0.47 $t_{38} = -1.5$, $P = 0.14$, cdlSNr: Pearson's $\rho = -0.08$, $P = 0.5$, value AUC = 0.18 $t_{47} = -11$, $P = 9e-16$, aversive AUC = 0.59 $t_{47} = 5.1$, $P = 5e-6$). Same format as Fig. 3. Source data are provided as a Source Data file.

Fig. 13) but we did not observe a significant difference in onsets between SNr and vlPFC across domains (Supplementary Fig. 14).

## Discussion

At the most basic level, objects ecological relevance can be viewed across three domains of appetitiveness, aversiveness and novelty. Previous work showed that stable reward history significantly changes the visual responses to objects in the corticobasal circuitry at both cortical[14,15] and basal ganglia level[8,26–28] including the basal ganglia output at cdlSNr. The value memory was found to be represented with high granularity and similar latency in both vlPFC and in cdlSNr[19], suggesting a common and distributed coding across the corticobasal circuitry. It was not known whether such observations can be generalized to the other domains such as novelty and aversiveness. To address this issue, neural responses across multiple domains were recorded within the same neurons in vlPFC and cdlSNr. Results showed that the same population of neurons in both vlPFC and cdlSNr encode aversive and appetitive domains. Importantly, the polarity of the

aversive signal was found to be related to the salience rather than the negative valence of aversive objects and was encoded in a correlated fashion with value signal in both regions (Figs. 2–6, Supplementary Figs. 4–7). Results showed that vlPFC neurons encoded value and novelty signals with a comparable size and in a correlated fashion while in cdlSNr the novelty signal was subdued compared to the value signal, had a delayed onset latency and was not correlated with the value signal in the population (Fig. 1, Supplementary Figs. 1, 13–14). These results show that past experiences with objects change passive visual responses to objects within the cortex and basal ganglia circuitry which are known to influence gaze (an action circuit) but with some differences across the nodes.

vlPFC was found to be sensitive to all three domains of object memory including reward, aversiveness, mere exposure (novelty/ familiarity). Importantly, these domains were not encoded by separate populations rather they were encoded by the same neurons and in a correlated fashion. The degree and sign of responding were consistent with the ecological salience of the

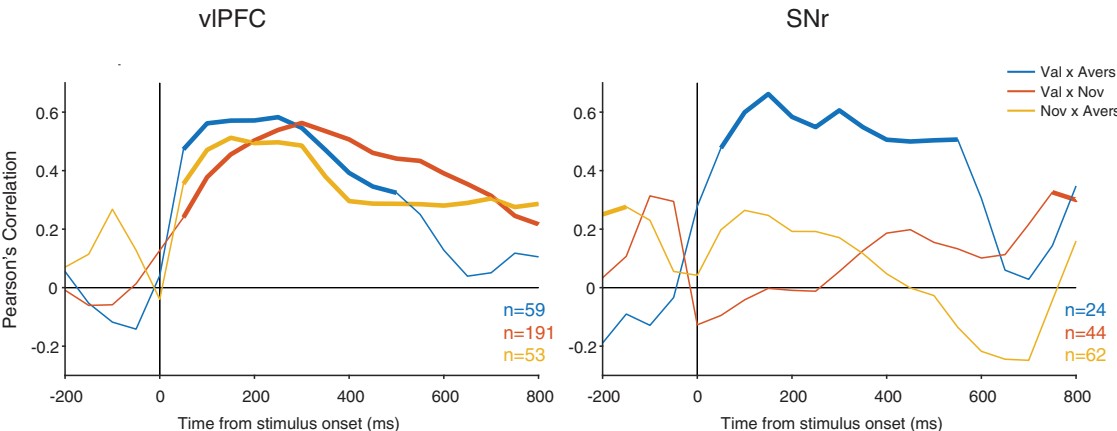

**Fig. 6 | Sliding pairwise correlation between value, novelty and aversive (airpuff) signals in vlPFC and cdlSNr.** Pairwise sliding correlation in 300 ms windows between value signal (good minus bad), novelty (novel minus familiar) and aversive signal (airpuff minus neutral) across neurons in vlPFC and cdlSNr across both monkeys. Think lines show significant correlations ($p < 0.05$). The pairwise correlation is calculated for neurons recorded with two set types with number of neurons noted. vlPFC encodes salience memory across all dimensions in a correlated manner while cdlSNr encodes salience memory only across value and aversive dimensions in a correlated manner. Val value (good/bad) sets, Nov novel/familiar sets, Avers Airpuff sets.

objects measured during free viewing (Supplementary Figs. 4, 6, 8). While in this study, we only considered long-term familiarity and absolute novelty, previous evidence showed the same vlPFC neurons to have significant suppression to recently viewed objects (relative novelty aka recency)[5,19]. Recency is also shown to reduce objects' ecological salience[5]. Together these results suggest vlPFC to encode a common currency signal related the attention worthiness of objects. This is consistent with the role of vlPFC in controlling gaze via anatomical projections to frontal eye field (FEF)[29] and superior colliculus (SC)[9].

cdlSNr showed significant coding of salient aversive objects similar to appetitive objects. On the other hand, the novelty response in cdlSNr was much weaker compared to value and aversive signals (Figs. 1–5) despite salience of novel objects (Supplementary Fig. 8). Furthermore, unlike vlPFC, the same cdlSNr neurons did not show any sensitivity to recency of objects (another form of perceptual salience) in a previous study[19]. These findings suggest cdlSNr to be sensitive to the outcome-related object salience rather than coding salience in its most general form. Interestingly, neighboring putative dopamine neurons to cdlSNr were found to have significant value signal but little novelty signal for objects (Supplementary Fig. 3). Lack of novelty coding in putative DA neurons may seem rather surprising. But despite the widespread belief about the role of DA in novelty processing, most previous studies on DA-ergic activity could not fully dissociate stimulus novelty from confounds such as recency/sensory surprise[30], reward expectation and learning[31,32] and movements such as orienting[33] and sniffing[34]. Furthermore, studies that manipulate novelty seeking with DA depletion, agonist and antagonists[35] in addition to being often confounded with concurrent changes in locomotion may not be relevant here since DA firing does not always correlate with DA release in the target area[36]. Other studies with controls for these confounds are based on fMRI signals in SNc which do not afford the resolution to infer DA neural firings[37]. Interestingly, a recent study with a well-designed design confirms this lack of object novelty signal in DA neurons as well as in the lateral habenula as one of the main inputs to DA neurons[38].

Additional studies may be required further verify lack of novelty coding in DA neurons.

These results along with previous electrophysiological and anatomical findings[9,19,24,27,29,39–45], help reveal the outlines of the circuit model involved in object salience memory (Fig. 7). Based on this model, vlPFC receives novelty and recency information from IT cortex[46] and value/aversiveness information from basal ganglia output, cdlSNr, via the medial nucleus of thalamus[43] consistent with its role as an information hub for mediating behavior in multifaceted and complex contexts[47–50]. Given that the cdlSNr receives indirect cortical input from vlPFC and temporal cortex[41,51,52] both of which are known to be sensitive to object novelty[10,24,46,53], the current findings suggest the intriguing possibility that the basal ganglia circuitry works to enhance value and aversive object memory signals while eliminating other sources such as novelty and recency[19] (Fig. 1, Table 1). Notably, the output of vlPFC to caudate tail is modulated by dopaminergic input which predominantly codes value and aversive information but not novelty[3,27,38], Supplementary Fig. 3) providing one mechanism for gating out sources of salience that are not outcome related. We note that given recent fMRI data[14] and anatomical evidence on targeting of IT cortex by cdlSNr[54] it is possible that IT cortex also encodes value/aversive object salience but such imaging evidence have to be verified by future electrophysiological recordings (thus not yet included in the circuit model). Recent evidence also suggests an indirect projection route from IT cortex to SC via Zona Incerta (ZI) which can carry novelty information[38]. ZI is also known to project to neocortex thus providing another potential route for novelty information to reach prefrontal cortex[55]. Finally, lateral intraparietal area (LIP) which is also known to encode object salience and has reciprocal connectivity with prefrontal cortex[56] is also found to be sensitive to novelty, value[22,24] and possibly aversive salience[20]. The exact role and interactions of this area in object salience memory needs to be further studied.

In real life, novel objects often carry outcome uncertainty with them. Given our previous report on coding of uncertain value memory in vlPFC and SNr, one interpretation of novelty responses seen in vlPFC can be that those novel objects signaled a possibility of reward. However, the lack of a strong population response in SNr to novel objects violates such value-related interpretations for novel objects especially given the strength of SNr responses when value memory is involved. Furthermore, in SNr a value-related interpretation predicts a higher excitation to familiar objects (which has zero reward) compared to bad objects (which had small reward) which is the opposite what is observed (Fig. 1d). Given these observations and the fact that in our

**Table 1 | Summary of Salience memory coding in vlPFC and SNr**

|       | Value | Aversive | Novelty | Recency |
|-------|-------|----------|---------|---------|
| vlPFC | ✓     | ✓        | ✓       | ✓       |
| SNr   | ✓     | ✓        | ✕       | ✕       |

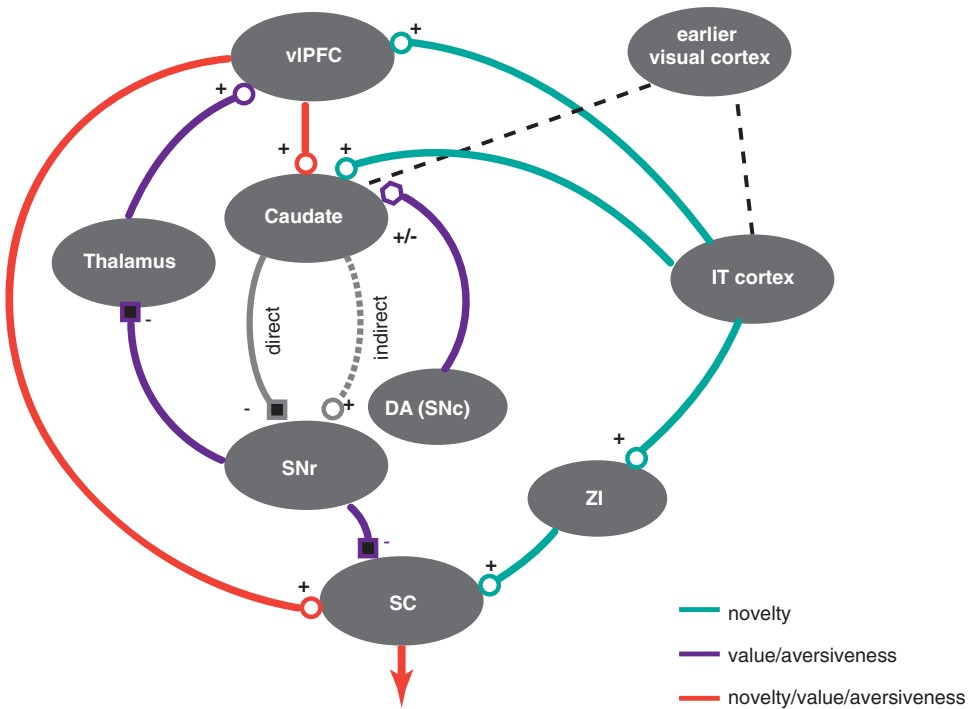

**Fig. 7 | Circuit model summary of salience memory flow across cortex, basal ganglia and thalamus.** The connectivity between prefrontal cortex, temporal cortex, basal ganglia and thalamus along with projection signs represented by shape of terminal are shown (circle: excitatory, square: inhibitory or pentagon: modulatory). Connections are color coded by the presumed type of information they carry (value, aversiveness and novelty). Direct and indirect pathways inside basal ganglia are noted as grey connections. Dopamine neurons residing in the

substantia nigra compacta (SNc) project to caudate carrying value and aversive signals. Prefrontal cortex and SNr have direct projections to SC and temporal cortex has indirect projection to SC via ZI to direct gaze and attention. ZI may also project directly to vlPFC providing novelty signals (not shown). Low level visual information for objects is supplied to both IT cortex and caudate by early visual cortex (shown by black lines).

paradigm novel objects seen in novel/familiar context were never used before or in the future, it is unlikely that the novelty responses observed can be reduced to reward or outcome uncertainty.

Why should there be some regions that are specialized for value and familiarity memory and some that combine both types of information ?[24]. To begin with, value and novelty have different origins and require different neural computations and thus can naturally arise in separate circuits. Furthermore, maintaining separate circuitry for value and novelty could be important in cases where decisions have to be made based on one aspect of object memory independent of the others. On the other hand, in cases where object salience regardless of origin is important, a node with access to a common salience signal such as vlPFC can control behavior more efficiently.

While the contrast between aversive and appetitive coding is often used to argue for valence vs salience (or valence vs arousal) coding across different brain regions[3,13,57], our measurement of actual learned salience across three different forms of punishing outcomes (airpuff, aversive taste and time-out) showed that despite their general negative valence, aversive object can vary on magnitude and even sign of salience. While airpuff associated objects tended to have higher salience than neutral objects (positive salience), saline and time-out object had similar or slightly lower salience compared to neutral objects (negative salience) (Figs. 2, 4, Supplementary Fig. 4, 6). Thus, without actual measurements, assertions about object salience in the aversive domain and the interpretation of neural responses may be ill-founded.

In summary, our results showed that consistent history of reward, punishment or novelty (i.e. lack of any history), affects passive visual responses to objects within a given neural population in both cortex and basal ganglia in ways that better correspond with the objects learned salience rather than their valence. Whether the memory of

these past object experiences is primarily stored within this cortico-basal loop or is stored elsewhere and relayed to this circuitry from an outside node is an outstanding question that requires further investigation. Furthermore, investigations of the neural mechanism that gives rise to the difference between vlPFC and cdlSNr with regard to their encoding of novelty signal and its behavioral significance may provide important clues about contrasting roles of cortex and basal ganglia in attentional control toward objects. Finally, given the role of vlPFC[48,58] and SNr[59] in controlling gaze, it remains to be seen whether and how the enhanced visual responses to salient objects translate to tasks requiring an actual gaze-orienting behavior.

## Methods

### Subjects and surgery

Two male adult rhesus monkeys (*Macaca mulatta*) were used in all tasks (monkeys B and R ages 7 and 10, respectively). All animal care and experimental procedures were approved by the National Eye Institute Animal Care and Use Committee and complied with the Public Health Service Policy on the humane care and use of laboratory animals. Both animals underwent surgery under general anesthesia for head holder and recording chambers installment on the head and scleral search coils for eye tracking in the eyes. The prefrontal recording chamber was tilted laterally and was placed over the left and right prefrontal cortex (PFC) for monkeys B and R, respectively (25° tilt for B and 35° tilt for R). The SNr chamber was tilted posteriorly and was centered on midline allowing access to cdlSNr on both hemisphere (40° tilt in both monkeys). Following confirmation of recording chamber position using MRI, craniotomies over PFC were performed during a second surgery. The craniotomies for cdlSNr were performed on the same hemisphere as the PFC recording chamber (left and right cdlSNr for monkeys B and R, respectively). In monkey R the craniotomy was

extended to allow recordings on the left cdlSNr as well. The recording was done through grids placed over the chamber with 1 mm spacing.

## Recording localization

Substantia nigra (SN) recording localization in both subjects were done using T1- and T2- weighted MRI (4.7 T, Bruker). T2- weighted MRI is especially useful for imaging SNr area due to higher iron content[60]. During imaging, the recoding chambers were equipped with a grid with 2 mm hole spacing and were filled with gadolinium for better contrast. The location of SN in each monkey was further verified using the standard monkey atlas (D99 atlas[61]) which was brought into each monkeys native space using the NMT toolbox[62] and the projection of SN as reachable through the posterior recording chamber was visualized and confirmed to coincide with the recording locations (refer to Supplementary Fig. 1 in[19]). vlPFC recordings were localized and reconstructed using the same scans per monkey and using the D99 atlas (refer to Supplementary Fig. 1 in[15]).

## Stimuli

Visual stimuli with fractal geometry were used as objects[63]. One fractal was composed of four point-symmetrical polygons that were overlaid around a common center such that smaller polygons were positioned more toward the front. The parameters that determined each polygon (size, edges, color, etc.) were chosen randomly. Fractal diameters were on average ~7° (ranging from 5°–10°). Monkeys saw many fractals across appetitive, aversive and perceptual domains. For the good/bad sets, monkey B and R saw 96 (12 sets) and 104 (13 sets) objects in good/bad sets (half good/half bad, Fig. 1a). For novel/familiar sets, both monkeys had 8 familiar objects. Monkey B saw 360 and monkey R saw 580 novel objects for these recordings (Fig. 1a). For aversive sets, each monkey B and R saw 3 sets for each airpuff, saline and timeout types (24 objects in total for each monkey Figs. 2, 4). Finally, for neural responses shown in Supplementary Fig. 9, data was from 4 amount and 4 probability sets in monkey B (40 objects in total) and 6 amount sets and 4 probability sets for monkey R (50 objects in total). Overall, monkey B saw 528 objects and monkey R saw 766 objects across all object types reported in this study.

## Task control and Neural recording

All behavioral tasks and recordings were controlled by a custom written VC++ based software (Blip; http://www.robilis.com/blip/). Data acquisition and output control was performed using National Instruments NI-PCIe 6353. During the experiment, head-fixed monkeys sat in a primate chair and viewed stimuli rear-projected on a screen in front of them (~30 cm) by an active-matrix liquid crystal display projector (PJ550, ViewSonic). Eye position was sampled at 1 kHz using scleral search coils. Diluted apple juice (33% and 66% for monkey B and R respectively) was used as reward.

Activity of single isolated neurons were recorded with acute penetrations of glass coated tungsten electrodes (Alpha-Omega, 250μm total thickness). The dura was punctured with a sharpened stainless-steel guide tube and the electrode was inserted into the brain through the guide tube by an oil-driven micromanipulator (MO-972, Narishige) until neural background or multiunit was encountered for vlPFC recording sessions or until firing pattern characteristics of SNr were observed in cdlSNr recording sessions after passing through cortical and subcortical structures guided by MRI. The electric signal from the electrode was amplified and filtered (2 Hz-10 kHz; BAK amplifier and pre-amps) and was digitized at 1 kHz. Neural spikes were isolated online using voltage-time discrimination windows. Spike shapes were digitized at 40 kHz and recorded for 4.5 ms (average shape of 300 spikes per neuron). An attempt was made to record all well-isolated and visually responsive neurons (visually response to neutral familiar fractals in passive viewing tasks or to flashing white dots in various locations). The results reported are from a total of 73

and 124 vlPFC neurons in monkeys B and R, respectively and 36 and 50 cdlSNr neurons in monkeys B and R, respectively and 3 and 6 dopaminergic neurons in monkeys B and R, respectively. Parts of the vlPFC and cdlSNr results were previously reported[15,19].

## Neural data analysis

Responses were time-locked to object-onset for analysis in the passive viewing task. The analysis epoch for calculation of AUCs was from 200–600 ms after object onset in passive viewing task. The discriminability based on learned values was measured from average firing during analysis epoch across trials using area under receiver operating characteristic curve (AUC). Wilcoxon rank-sum test was used for AUC significance for individual neurons (e.g. Fig. 1g, h). For onset detection the first response peak after object onset was detected using MATLAB *findpeaks* with a minimum peak height of 1.64 corresponding to 95% confidence interval. The onset was determined as the first valley before this peak (valleys within baseline) using *findpeaks* on the inverted response.

## Value Training: saccade task

Each session of training was performed with one set of fractals. The good/bad sets consisted of 8 fractals (4 good /4 bad fractals). Bad fractals were paired with low reward (0.07 ml and 0.11 ml in monkeys R and B, respectively) and good fractals were paired with high reward (0.21 ml and 0.35 ml for monkeys R and B, respectively). The different juice amount was customized for each monkey based on his water motivation and to ensure satisfactory cooperation. The high to low juice amount was about 3 to 1 in both subjects. In a given trial, after central fixation on a white dot (2°) one object appeared on the screen at one of the five peripheral locations (10–15° eccentricity) or at center. In some sessions, fractals were shown on 8 radial directions (45° divisions). After an overlap of 400 ms, the fixation dot disappeared and the animal was required to make a saccade to the fractal. After 500 ± 100 ms of fixating the fractal, a large or small reward was delivered. Diluted apple juice (33–66%) was used as reward. The displayed fractal was then turned off followed by an inter-trial intervals (ITI) of 1–1.5 s with a black screen. Breaking fixation or a premature saccade to fractal during overlap period resulted in an error tone (<7% of trials). A correct tone was played after a correct trial. Normally a training session consisted of 80 trials with objects presented in pseudo-random order. Each object set was trained for at least 5 sessions prior to test of long-term value memory. To check the behavioral learning of object values, choice trials with two objects with different reward association were included randomly in one out of five trials (20% of trials). During the choice the two fractals were shown in diametrically opposite locations and monkey was required to choose one by looking and holding gaze for 500 ± 100 ms on a fractal after which both fractals were turned off and the corresponding reward would be delivered. Only a single saccade was allowed in choice trials. The location and identity of fractals were randomized across choice trials. Choice rate was >99% in good/bad sets.

The amount and probability sets reported in Supplementary Fig. 9 were also trained using the saccade task. The amount sets consisted of 5 fractals with linearly increasing reward from low to high reward for each monkey. The probability sets consisted of 5 fractals associated probabilistically with low or high rewards but with a linearly increasing high reward probability. Note that in this case objects 1 and 5 in amount and probability sets had the exact same reward size as good and bad objects in good/bad sets, respectively and had no reward uncertainty. For amount and probability sets the average pairwise choice was 90% and 96%, respectively [for detailed results refer to[19]].

## Aversive training: Pavlovian task

We used a Pavlovian task to train aversive associations for the aversive sets. The main difference between this task and saccade task was that

subjects were not required to fixate fractals to receive outcome during the Pavlovian task as some objects were aversive and thus it was impractical to ask monkey to fixate them to receive punishment. Here animals received the outcome for each object after a single object appeared in the screen (force trials which happened in about 25–50% of trials). To offset the effect of punishment and motivate the animal to cooperate for the aversive sets, the rewarding objects had 1.5 times the large reward amount for good/bad sets. All trials were preceded by an inter-trial interval (ITI) of $1250 \pm 250$ ms with a blank screen. After this, a fractal was shown randomly at one of 5 locations (4 radial, eccentricity 15 and at center) for $500 \pm 100$ ms. Afterwards, the fractal was extinguished and the corresponding outcome was delivered. Neutral objects did not lead to any outcome. Good objects resulted in a large reward. Airpuff objects resulted in 100 ms of 8-psi airpuff delivered to the right eye. The puff spout was positioned ~5 cm away from the temporal side of the eye. Saline objects resulted in delivery of normal saline (0.07 ml and 0.11 ml in monkeys R and B, respectively). Time-out objects resulted in an 8 s delay to start of the ITI. The ITI for time-out set was reduced to $750 \pm 250$ ms to enhance impact of this 8 s time-out. All aversive sets consisted of 8 fractals with 4 neutral, 2 aversive, and 2 rewarding (good)objects. About 50-75% trials in the Pavlovian task, were choice trials with the same temporal structure as choice trials in the saccade tasks but with the difference that here multiple saccades were allowed before committing to an object.

### Visual response: passive viewing task

A passive viewing trial started after central fixation on a white dot (2°). The animal was required to hold a central fixation while objects from a given set were displayed randomly with 400 ms on and 400 ms off schedule (Fig. 1b). Animal was rewarded for continued fixation after a random number of 2–4 objects were shown. A trial would abort if animal broke fixation or made a saccade to objects (<1% of trials). Objects were shown close to the location with maximal visual response for each neuron as determined by receptive field mapping task (for detail description and analysis of this task refer to[19]. When this maximal location was close to center (<5°) passive viewing was sometimes done by showing objects at the center. A block of passive viewing consisted of 5-6 presentations per object. In most cases, more than one block was acquired for a given set.

### Behavioral salience memory: free-viewing task

Each free-viewing session consisted of 15 trials with fractals from a given set. In any given trial, four fractals would be randomly chosen from the set and shown in one of the four corners of an imaginary diamond or square around center (15° from display center, Fig. 2c). Fractals were displayed for 3 s during which the subjects could look at (or ignore) the displayed fractals. There was no behavioral outcome for free viewing behavior. After 3 s of viewing, the fractals disappeared. After a delay of 0.5–0.7 s, a white fixation dot appeared in a random location on the screen (center or eight radial directions). Monkeys were rewarded for fixating the fixation dot. This reward was not contingent on free viewing behavior. Each trial was followed by an ITI of 1–1.5 s with a black screen. Behavioral data for good/bad, novel/familiar and aversive sets is from monkeys B and R used in electrophysiology. Behavioral data for amount and probability sets are from four monkeys B, R, D and U who did 12, 18, 6 and 7 sessions with probability sets, respectively and 12,12,7 and 7 sessions with amount sets, respectively.

### Free-viewing analysis

Gaze locations were analyzed using custom written MATALB functions in an automated fashion and saccades (displacements >2.5°) vs stationary periods were separated in each trial[5,64]. Percentage of first saccade to a given object following the display onset was used as the measure object salience (chance level: 25% in our free viewing with 4 objects).

### Statistical tests and significance levels

One-way and two-way ANOVAs were used to test main effects and interactions for neural responses and for behavior Error-bars in all plots show standard error of the mean (s.e.m). All post-hoc tests were Tukey's hsd. Significance threshold for all tests in this study was $p < 0.05$. *ns*: not significant, $*p < 0.05$, $**p < 0.01$, $***p < 0.001$ (two-sided).

### Reporting summary

Further information on research design is available in the Nature Research Reporting Summary linked to this article.

## Data availability

All data needed to evaluate the conclusions in the paper are present in the paper and/or the Supplementary Materials. The neural data from vlPFC and SNr are provided in Supplementary Data 1 and the behavioral choice and free viewing data reported in this study are provided in Source Data file. Source data are provided with this paper.

## Code availability

Standard or available matlab functions such as hist, findpeaks and scoreAUC are used for PSTH, onset detection and neuronal AUC. All codes can be made available upon request.

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

## Acknowledgements

This research was supported by the Intramural Research Program at the NIH, National Eye Institute (grant no. EY000415-15) made to OH.

## Author contributions

A.G. and O.H. designed the experiment. A.G. collected and analyzed the data. A.G. and O.H. wrote the paper.

## Competing interests

The authors declare no competing interests.
