## [Peer Review File · Nature Communications]

Saliency Memories Formed by Value, Novelty and Aversiveness Jointly Shape Object Responses in the Prefrontal Cortex and Basal GangliaReviewers' comments:

Reviewer #1 (Remarks to the Author):

In this study Ghazizadeh and Hikosaka characterize the firing rate response and the correlation of firing of neurons in prefrontal cortex and substantia nigra to passively viewed stimuli that were novel or perceptually familiar or associated with Pavlovian high or low reward outcomes, aversive airpuffs time outs or bitter taste.

The authors find that vLPFC neurons responded stronger to stimuli that were salient given prior learned appetitive and aversive associations or by being unfamiliar (novel). This vLPFC response patterns was distinct to neuronal firing in the substantia nigra which did contain less novelty related responses but responded to objects with prior learned salient rewarding and aversive outcomes with suppressed and increased responses. The saliency related responses in both vLPFC and SN were correlated across the neural populations. The authors interpret this correlation as indicating that the same neurons encode appetitive and aversive value based saliency, which in vLPFC also includes novelty responses suggesting the vLPFC codes for cross-domain object saliency.

Overall the findings of the study are important and interesting, but they lack statistically sensitive methods to distinguish the degree of neural coding of individual saliency domains versus a common 'cross-domain' saliency. Moreover, the manuscript does not consider a larger literature on novelty and saliency coding in the introduction and lacks the discussion of the latencies of saliency coding amongst others. It contains multiple grammatical errors.

Aspects that deserve consideration to clarify the results and help readers understand and gain confidence in the main findings include the following:

1) It is not clear how much the neural responses distinguish different unique objects and whether or how much the learned outcome associations of the objects or the novelty enhances the discrimination of the objects.

The current manuscript implicitly assumes that neurons encode learned associations or novelty. The question whether neurons distinguish objects and how much the object code is enhanced by higher value, by different aversive outcomes, and novelty remains unaddressed.

One possible way to address this point would be a decoding analysis that clarifies how accurate the salience category can be predicted by the firing rate response. This approach could also quantify whether appetitive and aversive outcomes can be decoded separately better than a combined common-currency saliency category.

2) It is unclear whether firing increases and firing decreases of cdISNr neurons are both informative and what their origins are. Do both response types increase discrimination of the learned object values? An explicit discussion of this aspect of the results is missing in the current manuscript.

3) The authors provide time resolved average analysis results. But there is no statistical latency analysis provided that could clarify when in time the population of salience-/value-/aversive- outcome coding neurons show a differential firing increase. Similarly, does the onset latency about the learned salience / value / novelty response significantly differ between vIPFC and cdSNr ? The discussion mentions other brain areas coding for similar saliency, but there is so far no discussion on how the latencies found in this study compare to those reported in other brain areas reported in other studies.

4) How are saccadic response initiation and execution affecting firing responses of neurons and does saccade aligned activity distinguish aversive, appetitive, or novel objects. It is not clear why the gaze - orienting responses are not described given that the study speaks directly to the influence of different saliency domains to control gaze ? Wouldn't saccade onset aligned responses be informative to infer saliency based gaze control

Can the authors also clarify why the analysis window (up to 600ms) overlapped with the saccade triggering signals (fixation stim offset after 400ms) and whether the choice of the temporal window affects the main results (it does not seem so , but the error bars are not shown).

5) Related to the previous point, it is not clear how stable the gaze was following object onset. The assumption is that microsaccade rate does not differ between object categories but this is not shown. An analysis of gaze in different object categories is needed to be confident that the results are not accounted for by different orienting response triggered by the different saliency domains. This is particularly important considering that aversive outcome associations trigger fast automatic gaze shifts.

6) Some results are described in a confusing way: Eg p 6 describes that the " the aversive signal and value signal were positively correlated in both vIPFC and cdISNr" which suggests that the response are reflecting learned stimulus saliency. But then the authors say that "the same neurons ... encode object value memory", which is the opposite of saliency coding. It remains unclear how many neurons are actually coding for the learned saliency or the reward value and how these coding schemas are best statistically distinguished.

The lack of a clear statistical approach to disentangle positive value, aversive responses and saliency coding is becoming apparent when the authors find a positive correlation of pos. reward ('value') related responses and aversive responses in the cdSNr, which is interpreted as reflecting an "outcome related saliency" signal. Exemplar raster plots for the different valence conditions would help appreciating the saliency response

7) The abstract is not clear. There are multiple grammatical errors. There are concepts used that lack clarity like "salience memory", and should be described/introduced more explicitly.

8) The results section lacks statistical significance values. It is written in a descriptive style with multiple assertions like "...neurons on average showed stronger excitation ..." having no p value and test specified that would support the statement or "...aversive signal and value signal were positively correlated in both vIPFC and cdLSNr ..." without providing the correlation coefficient and significance value. This issue is pervasive throughout the results section and is not ameliorated by the statistical results accompanying some figures as many statements in the text are made without explicit link to a figure panel. To gain confidence in the results and understand its reliability it will help to have statistical data for each assertion and an explicitly mentioning of which test statistics has been used for the inferential statistics.

9) Related to the previous point, the authors may want to provide a direct statistical comparison of vIPFC versus cdLSNr findings whenever the difference between areas is interpreted.

10) Figures (Fig. 1C,2E) of average firing rates are presented without error bars / error shading, which is not appropriate as it does not allow understanding how typical the average (shown) response is. Error bars should be included. Similarly, the text provides at multiple times average values (e.g. percentage difference in firing) without giving standard errors/standard deviations across the neural populations. This should be added.

11) The expectation was that the introduction introduces the literature on how appetitive, aversive, novelty information is encoded in vIPFC and how this differs to the basal ganglia. This literature is missing and is not provided. A rich existing literature is not mentioned.

12) The result section does lack specific basic information:

- p2 line 37: how many sessions in each monkey?

- p3, line 1: how many fractal objects were used in each category on average?

- p 3 line 19L how many previous trials or encounters were done for familiar objects (on average \pm SE) ? Across how many sessions on average (the methods only states >10 session per set). This information is needed to understand how familiarity and long term saliency memory is defined.

minor

abstract:

- shows or has shown instead of '...evidence show...'

- "visual responses within neurons" ... 'of neurons'

- line 20

the same or different population of neurons

- p. 7 "... two dimensions (value, aversiveness, novelty) ..." 2 or 3 dimensions?

there are multiple grammatical errors that are not listed individually here. The errors continue in the methods section ...e.g. is in "T2 relaxation times in much shorter in SNr area" or singular in "a scleral search coils",...

Reviewer #2 (Remarks to the Author):

This is an interesting study that sheds new light on the role of the vIPFC and the SNr in maintaining the memory of value novelty and aversiveness. However, in my opinion, the Authors should show additional evidence in order to support their hypothesis.

Major Points

The description of the training tasks, which is clear in their previous paper (Ghazizadeh A., et al, 2021), here is confusing and difficult to understand. The Authors should rephrase the description of the tasks in the Method section.

The interpretation of the passive viewing tasks relies on the behavior in the training tasks. However, analysis of the behaviors is not sufficient. For instance, the report only the results of the choice made by the monkeys in choice and in the free viewing tasks. As discussed below, the analysis of the saccade latency is a good measure to establish the behavioral relevance of an object.

In a previous study (Ghazizadeh A., et al, 2021) the authors found a strong memory value and an amount and probability reward memory signals in both areas. They also found a stronger activity for objects with uncertain reward probability compared to objects with 0 and high reward probability in vIPFC and a gradually stronger inhibition with the increasing reward amount and expectation and a slight excitation for small reward amount and for 0 reward probability in SNr (Figure 5 A- D of Ghazizadeh A., et al, 2021). Here there are 4 types of objects associated with 0 rewards. The novel and the familiar objects (the task of figure 1) and the neutral objects (figure 2). In both the familiar and the neutral objects trials, the probability of reward is zero, while in novel object trials, there may be uncertainty about the reward. Isn't possible that the familiar and novel objects are instead associated with expected reward?

The neurophysiological data in this study may support the hypothesis. In this study, the neural response of vIPFC neurons to the familiar objects is lower than the response to novel objects, as the response to 0 reward probability vs the response to the uncertain reward in their previous study (Ghazizadeh A., et al, 2021). Likewise, neurons in SNr respond with an increase of activity to familiar and novel objects, as in the low amount and low reward probability. The same observations can be drawn from Figure 1E of both vIPFC and SNr. In vIPFC, the higher the activity of the Bp neurons, the higher the activity for the familiar objects. In other words, the results might be explained by the gradual increase of the response with the expected reward amount and probability: familiar (no reward), uncertainty (novel stimulus), low reward, and good reward. The authors should clarify why the familiar and novel objects do not have any expected (low or uncertain) reward value and report the latencies of the saccade to good, bad, familiar, and novel in the choice tasks.

Figure 1D shows that in SNr, the onset of the novelty signal is significantly later than the onset of the value signal. However, in the supplementary figure (S2 that is S1) it seems that only in monkey B the onset time of novelty signals was later than the value signal. If the onset of the novelty is later than the value only in monkey B, the authors should report it.

I found the section regarding the results of aversive signals muddled and difficult to read.

From what I understood, the choice task is a decision-making task, in which the monkeys can choose only one of two objects, while in free viewing there is no such constrain because monkeys are not required to make any choice.

The authors should report the latencies of the first saccades in free viewing and choice tasks in a supplementary figure and compare the latencies of the first saccades to good objects with the latency of the first saccade to aversive and neutral objects. If there is not a significant difference, I would expect that in the choice task a significantly longer latency of saccades to good objects when paired with aversive than when paired with neutral objects since in the choice trials there is a strong competition between the two objects. The authors should plot the difference between the response to good objects and the response to aversive objects in the value vs aversive signals plot (Fig 1E) to better compare the behavior with the neural activity in vIPFC and SNr, separately for the two monkeys.

Minor points

In this study, the visual response was evaluated in a time window from 200 to 600 ms after stimulus onset, while in the previous study it was calculated from 100 to 400 ms after the object. Is there any particular reason why they choose two different epoch windows?

Fig S5: the range of the blinking rate should be the same (0 to 0.6) for both monkeys

There are two supplementary figures 2.

The way the subplots in each figure are named is confusing and difficult to follow. For instance, the subplots 1C of figure 1, should be broken into two: figure 1 C (vIPFC) and 1 D (SNr). The same for figure 1 D and 1 E, and for figures 2 D and 2 E.

Reviewer #3 (Remarks to the Author):

Here, the authors present a paper that extends their work on object-based memories stored in cortico-basal ganglia circuits to include objects that are not reward associated. I think the authors' intents here are noble, and I believe the experiments will lead to important insights, but in its current form I have several major concerns about the conceptualization and implementation of the research described in this paper. The major concerns revolve around the conceptualization of salience and value, how we measure these concepts, and on the amount of weight that is attributed to the results from monkey R. To the authors, the neural data supports 'salience' coding, rather than value coding. However, the lack of clear definitions, the crude measurement of value, and the correspondence between the behavioral and neural data from monkey R regarding the airpuff raises serious challenges to their interpretation. Accordingly, this is valuable data to show but the interpretations should be strongly moderated. In detail:

1) A clear definition of 'salience' is lacking from this paper. Elsewhere, I have read about visual salience, attentional salience, physical salience, motivational salience, emotional salience, psychological salience; the list goes on and on. These phrases do not all mean the same thing. For example, a loud bang caused by a passing car might be physically and attentionally salient, to a pedestrian, but it is not likely to be motivationally salient (Or maybe it is! The point is these concepts are separable). This lab drives the salience narrative in cortico-basal ganglia neurophysiology, and yet it is unclear what they mean by 'salience'. In figures 1-3, I inferred they are talking about motivational salience. Then, in figures 4 and 5, and in the associated text, they switch from using airpuff as punishment to using bitter tastes and time-outs. They write, on Line 18, page 6, '[t]o further test this hypothesis, object punishment history was created by using aversive outcomes (aversive taste or time-out) that did not change object salience (Fig 4A).' This suggests that they do not mean motivational salience, but rather physical or attentional salience. (To reinforce that interpretation, they go on to show that the animals avoid cues paired with aversive tastes and time-outs, a behavior consistent with increased motivational salience for cues associated with those objects). After reading figures 4 and 6, I looked back at figures 1-3 and try to re-interpret them regarding physical or attentional salience, but it doesn't fit. I am left with the impression, frankly, that the amorphous definition of salience is required to make the 'salience theory' work. It would be much better if in the introduction the authors created a working definition of salience and apply it consistently throughout the manuscript.

2) Choice and value are related but not equal. Choices (here) are binary, whereas value is not. Choices are equal to value passed through a softmax function, or some other kind of thresholding operation. This poses a major challenge to scientists that want to measure value. Here, the authors argue that the neural responses which they recorded reflect salience rather than value, largely because they look more like the free viewing behavior than the choices. However, if we agree that choice and value are not equal, and we should because it is obvious, then the clear interpretation of this data as coding salience rather than value is hard to substantiate. Furthermore, the differences between monkeys B and R regarding the airpuff are interesting, and I believe they speak to this point. The lower left-hand panel for the cdlStr responses from B and R, in Figure 2E, perfectly correspond to the behavioral differences between the two monkeys. These neural differences are entirely consistent with a subjective value signal. The marginal distributions shown in Figure 3 and 5 are also consistent with subjective value, the only difference is a positive code in the PFC, and an inverted code in SNR.

3) The authors claim that the 'salience theory' explains 'peculiarities in neural responses' when risk is involved (page 6, line 35 and Fig. S6), but they do not provide enough details about those experiments to back up this claim. If the monkeys are risk seeking – and nearly all monkeys are risk-seeking for small rewards – then subjective values and risk attitudes explain S6 as well as the 'salience theory'. Here, as elsewhere in this manuscript, the population PFC codes value in a positive way (good higher than bad) and the population cdlStr codes value in an inverted fashion (bad higher than good). Furthermore, displaying these signals as a difference, as in Figure S6c, makes these neural responses look like the uncertainty responses the senior author showed in the septum (Monosov and Hikosaka, NatNeuro 2013 – one of my favorite papers of all time!). However, they are not pure uncertainty signals in the way the septum signals are. These responses are consistent with subjective value, not pure uncertainty.

4) Page 5, Line 35 is very confusing. I can't understand what they meant to write, so I will just leave this for the authors to address.

5) The circuit cartoon in figure 7 does not include a role for dopamine responses in novelty processing, and this omission is not consistent with many previous studies that have shown very strong novelty responses in dopamine neurons, and a role for those novelty responses in preventing latent inhibition. Here, if I read figure S2 correctly, the authors recorded 9 dopamine neurons. Can the authors explain why they see less dopamine novelty responding than previous studies? Is there a reason the field should re-assess the widespread understanding that novelty drives dopamine responding?

Minor

1) In previous papers, this lab has used 40psi stimuli, but there is no indication of airpuff parameters used here, that I can see.

2) Page 2 line 29 'paralleled measured' seems like a typo. There are many other places where the writing could be improved.

Dear Colleagues,

We would like to thank the editor for organizing the reviews and the reviewers for their many insightful and interesting comments. As a result of these comments, we have now revised the main text of the manuscript and updated a few of the figures (major additions to the main text are highlighted in yellow). Furthermore, we have done additional analysis and added 6 new supplemental figures (and combined two supp figures into one to keep the count under control). We are also presenting three figures in the response to reviewers which we did not include in the paper (Fig R1-3). Please find our point-by-point responses to each comment below.

Regards,
Ali Ghazizadeh and Okihide Hikosaka

Reviewers' comments:

Reviewer #1 (Remarks to the Author):

In this study Ghazizadeh and Hikosaka characterize the firing rate response and the correlation of firing of neurons in prefrontal cortex and substantia nigra to passively viewed stimuli that were novel or perceptually familiar or associated with Pavlovian high or low reward outcomes, aversive airpuffs time outs or bitter taste.

The authors find that vIPFC neurons responded stronger to stimuli that were salient given prior learned appetitive and aversive associations or by being unfamiliar (novel). This vLPFC response patterns was distinct to neuronal firing in the substantia nigra which did contain less novelty related responses but responded to objects with prior learned salient rewarding and aversive outcomes with suppressed and increased responses. The saliency related responses in both vIPFC and SN were correlated across the neural populations. The authors interpret this correlation as indicating that the same neurons encode appetitive and aversive value based saliency, which in vIPFC also includes novelty responses suggesting the vIPFC codes for cross-domain object saliency.

Overall the findings of the study are important and interesting, but they lack statistically sensitive methods to distinguish the degree of neural coding of individual saliency domains versus a common 'cross-domain' saliency. Moreover, the manuscript does not consider a larger literature on novelty and saliency coding in the introduction and lacks the discussion of the latencies of saliency coding amongst others. It contains multiple grammatical errors.

We thank reviewer for important comments. We would like to note that most statistical tests in this paper are provided in the figure captions but we have added additional test where noted. Please see our point-by-point responses and modifications including additional literature review and discussion of latencies in results below.

Aspects that deserve consideration to clarify the results and help readers understand and gain confidence in the main findings include the following:

1) It is not clear how much the neural responses distinguish different unique objects and whether or how much the learned outcome associations of the objects or the novelty enhances the discrimination of the objects. The current manuscript implicitly assumes that neurons encode learned associations or novelty. The question whether neurons distinguish objects and how much the object code is enhanced by higher value, by different aversive outcomes, and novelty remains unaddressed.

One possible way to address this point would be a decoding analysis that clarifies how accurate the salience category can be predicted by the firing rate response. This approach could also quantify whether appetitive and aversive outcomes can be decoded separately better than a combined common-currency saliency category.

We believe that the reviewer is raising two different points:

1) The first seem to be related to object selectivity in SNr and PFC and the possibility of having different selectivity for object within each category. We have partially addressed this issue for rewarding stimuli (within good and bad) in our previous two publications where we have looked at object selectivity and its change by value in vIPFC and SNr in detail (Science Adv 2021 Fig S5 and Cur Biology 2018 Fig 4C, S5). In brief, the object selectivity of vIPFC and SNr seem to be mostly the same for good and bad objects. To address the reviewer's point and extending our analysis to novel and aversive objects, we have repeated our previous analysis on these new objects using absolute pairwise AUC (which is a standard decoding tool) and have presented the results in supplementary Fig S10. Results show that object selectivity between good/bad, novel/familiar and aversive/good object are almost the same in both vIPFC and SNr. There was some trend for higher object discriminability for neutral objects in aversive sets which may be harder to interpret since aversive and good objects had two fractals and neutral had four fractals in each group thus AUC comparisons between neutral and others might not be the most efficient here.

The main findings are described in page 7 in Result section:

"Results also showed that object discriminability was mostly similar between main object categories across neuron types (good vs bad, familiar vs novel and aversive vs good objects) in both vIPFC and SNr (Fig S10, some exceptions in SNr: in aversive sets trending lower discriminability of neutral objects in Bp neurons and higher discriminability of good objects in NS neurons. In vIPFC: in good/bad and novel/familiar sets somewhat higher discriminability of good and novel objects in Gp neurons). This suggests that previous experience in appetitive, aversive or perceptual domains had modest if any effects on changing object selectivity within an object category in vIPFC and SNr which in any event were previously found to have low object selectivity by measures such as sparsity (15) or nonuniformity (19)"

2) The second suggestion involves using a decoding technique to predict object salience categories from firing rate. This point is already addressed for a given set by using AUC on value, novelty and aversive signals. Indeed, AUC is a standard decoding tool for an ideal observer. Our results show that by looking at firing one can significantly distinguish good from bad, novel from familiar and aversive from neutral or good depending on their salience (Figs 2-4). However, doing a decoding across set types and across all categories (e.g. novel vs aversive) in each region requires a few developments: While free viewing (FW) is giving us a handle to quantify salience of all categories on a single measurement scale (gaze bias) and we have seen a monotonic relationship between FW gaze bias and firing rate in vIPFC and SNr, the actual transformation between salience and firing rate on a global scale is not known. Indeed, in our previous work we have observed curious differences on firing to objects depending on what other objects they were trained with (set type dependances, see Science Adv 2021, Fig 5C). It is not currently known whether salience is a global metric or is rescaled for each set of objects.

While we believe the current data is strong evidence in favor of salience coding in both regions details such as the one discussed above have to be addressed before one can use firing to tell salience type of object across all three dimensions and across object sets examined here (value, novelty and aversiveness).

2) It is unclear whether firing increases and firing decreases of cdISNr neurons are both informative and what their origins are. Do both response types increase discrimination of the learned object values? An explicit discussion of this aspect of the results is missing in the current manuscript.

A thorough analysis of cdISNr neurons including the significance of both excitations and inhibitions in coding information and their possible origins and effects on saccade in downstream areas such as superior colliculus is done in previous publications (Yasuda et al, J neuroscience, 2013, Yasuda and Hikosaka, Journal of neurophysiology 2014, Amita et al Nat Comm, 2020, Ghazizadeh et al Science Adv 2021 Fig 8).

As for learning, the reviewer raises an important point. Obviously at the end of learning both Gp and Bp neurons in SNr discriminate objects based on their values as shown by significant AUC in Fig 1H. We have some unpublished recordings during the learning process in the SNr. The data shows that both firing rate increases to bad and decreases to good objects appear during the learning. We are including the results here for the reviewer in Figure R1 but since we are preparing a separate manuscript comparing the learning dynamics in vIPFC and SNr, we prefer to present this figure in that paper. The learning results in vLPFC is already published in Ghazizadeh et al, Curr biology 2018 Fig 2.

Learning in SNr

Figure R1: Learning in SNr. A) PSTH for response to good and bad objects in value learning saccade task averaged across chunks of 5 trials from left to right B) Trial by trial changes in firing rate (averages 100-400ms after object onset) to good and bad objects. Results show both excitation and inhibition to develop during learning. Note that the response to objects in saccade task is inhibitory from the beginning. The inhibition deepens for good objects while for bad objects inhibition becomes more shallow until turning to excitation.

3) The authors provide time resolved average analysis results. But there is no statistical latency analysis provided that could clarify when in time the population of salience-/value-/aversive- outcome coding neurons show a differential firing increase. Similarly, does the onset latency about the learned salience / value / novelty response significantly differ between vlPFC and cdSNr ? The discussion mentions other brain areas coding for similar saliency, but there is so far no discussion on how the latencies found in this study compare to those reported in other brain areas reported in other studies.

We have already shown latency comparisons for value and novelty in each region in Fig 1E-F and provided statistical reports (stats of the paper are in the figure captions). To address the reviewer's comment, we have now added two supplemental figures (Figs S13, S14). Figure S13 shows the latency comparison between value, novelty and aversive signals (separately for airpuff, saline and time-out objects) within neurons in vlPFC and SNr while Figure S14 shows comparison of value, novelty and aversive signals between vlPFC and SNr

These results are now described in page 8 lines 16-18 and stats are reported in Fig S13, S14 captions.

“Furthermore, results showed that in SNr the value signal seemed to have a faster onset compared to other domains such as novelty or aversive signals (Fig 1F, S13) but we did not observe a significant difference in onsets between SNr and vlPFC across domains (Fig S14).”

4) How are saccadic response initiation and execution affecting firing responses of neurons and does saccade aligned activity distinguish aversive, appetitive, or novel objects. It is not clear why the gaze - orienting responses are not described given that the study speaks directly to the influence of different saliency domains to control gaze ? Wouldn't saccade onset aligned responses be informative to infer saliency based gaze control

The reviewer raises an important point. Attentional bias toward a salient object is often concurrent with a gaze-orienting response (overt attention). Indeed, we are making extensive use of such gaze bias in free viewing to verify (and not just presume) salience for various object categories. Such verification has proven invaluable for example in the case of aversive objects where we found bias toward airpuff associated objects but not saline or time-out associated objects. This was important since most previous studies assumed salience for aversive taste or time-out objects without explicit testing (e.g. Roitman et al 2005, Leathers et al 2012).

However, while in almost all previous studies the responses to putative salient objects were measured during an active task (with reward or punishing outcomes delivered on each trial of object presentation), in this paper we are interested on how past experience changes the very visual responses to objects in the absence of any outcome (aka 'salience memory' which is created by past experiences and outcomes which could be irrelevant of actions and outcomes for the current task) .Therefore, all reported neural responses in the paper are done in 'passive viewing' where there is no outcome expectations nor any saccades to the objects.

Nevertheless, there is extensive literature that implicates both vIPFC (e.g. Roesch & Olson 2003, Bichot et al 2015)and SNr (e.g. Hikosaka & Wurtz 1985) to gaze control.

Given the interest of the reviewer in the firing rate during tasks with active saccades, here we present some data in-preparation for publication that involves making saccades to salient objects (see Fig R2 as an example for average vIPFC response in a different task we call 'free-looking' with saccades to good/bad or novel/familiar objects but still in the absence of outcomes.

Figure R2: vLPFC neurons show stronger excitation when saccading to good and novel objects. A) Free-looking task: This task consisted of consequent flashes of 2-4 objects randomly chosen from good/bad or novel/familiar fractals. Unlike passive viewing task, here during object display central fixation was turned off and the animal could make a saccade to an object if it so chose. Making or not making saccade had no consequence or outcome and the animal had to fixate the reappearing fixation dot at the object offset to continue the trial. Animals were rewarded at random intervals for fixating the central dot. B) Percentage of saccades to good was significantly higher (left) and saccades initiated faster (right) than bad objects. C) Percentage of saccades to novel was significantly higher (left) and saccades initiated faster (right) than familiar objects. D) Firing rate of population vLPFC neurons time locked to saccade was higher when saccade was toward good compared to bad objects (insets show average firing in [-150,150] ms window around saccade time. E) Firing rate of population vLPFC neurons time locked to saccade was higher when saccade was toward novel compared to familiar objects (inset format same as D)

Since in the current paper we are focused on the pure visual responses based on salience memory and addressing the saccade question requires recording both regions across the three domains with new tasks (such as free looking) we would like to address it in a separate paper. However, to acknowledge the point raised by the reviewer, we have added this as a discussion of future work in page 11 line 23-25:

“Finally, given the role of vIPFC (49, 60) and SNr (61) in controlling gaze, it remains to be seen whether and how the enhanced visual responses to salient objects translate to tasks requiring gaze-orienting behavior and affect the saccadic responses toward objects in this corticobasal circuitry.”

Can the authors also clarify why the analysis window (up to 600ms) overlapped with the saccade triggering signals (fixation stim offset after 400ms) and whether the choice of the temporal window affects the main results (it does not seem so, but the error bars are not shown).

Once again, all results presented were from passive viewing behavior where there is NO saccade. Animal simply fixates the central dot while multiple objects (2-4) are flashed in the neurons RF with 400ms on and 400ms off between the two consecutive stimuli and the fixation is never turned off during about of stimuli until after the last stimuli where we give monkey a reward. This point is now emphasized in the result section page 3 line 19-22:

“To test whether the memory of these past experiences with objects is reflected in the visual responses of the corticobasal circuitry, neural responses to objects were recorded using a passive viewing procedure in the absence of any outcome for objects and in the absence of saccades (Methods) (19).”

The choice of window does not affect our main conclusions but choosing a window until 600ms allowed for a better dynamic range for AUC in SNr to be used for correlations across domains. please see our extended answer to a similar point raised by reviewer 2 (first minor point). To verify this we have presented our results using [100-400]ms in Fig R3.

5) Related to the previous point, it is not clear how stable the gaze was following object onset. The assumption is that microsaccade rate does not differ between object categories but this is not shown. An analysis of gaze in different object categories is needed to be confident that the results are not accounted for by different orienting response triggered by the different saliency domains. This is particularly important considering that aversive outcome associations trigger fast automatic gaze shifts.

As mentioned previously all recordings are done in passive viewing with which animals had extensive (a few years of) experience and breaking fixation was rare across sessions in both subjects (<1% of trials). On reason exactly being that we wanted to make sure our responses are not affected by preparatory saccade activity and reflect purely visual signal that are affected by past memories. Furthermore, previously we have examined and found no evidence for mini saccades toward objects in the passive viewing (Ghazizadeh et al, 2018 PNAS, Fig S1B).

6) Some results are described in a confusing way: Eg p 6 describes that the " the aversive signal and value signal were positively correlated in both vIPFC and cdISNr" which suggests that the response are reflecting learned stimulus salience. But then the authors say that "the same neurons ... encode object value memory", which is the opposite of salience coding. It remains unclear how many neurons are actually coding for the learned salience or the reward value and how these coding schemas are best statistically distinguished.

In the sentence the reviewer is referring to we are saying that many neurons that are encoding the aversive signal (airpuff minus neutral) are also encoding the value signal (good minus neutral) and they do so in a correlated fashion (i.e. the stronger the aversive signal the stronger the value signal hence the significant positive correlation). This is not the opposite of salience coding but the very essence of salience coding since both airpuff and good stimuli are shown to induce positive gaze bias in free viewing despite their opposite valences.

Once more we would like to emphasize that we refer to these signals as aversive memory, value memory and salience memory since recordings are done in passive viewing of objects where response differences presumably reflect the past experience with objects (given the large number of objects and random assignments to groups, low level features are not expected to play a role here).

The lack of a clear statistical approach to disentangle positive value, aversive responses and salience coding is becoming apparent when the authors find a positive correlation of pos. reward ('value') related responses and aversive responses in the cdSNr, which is interpreted as reflecting an "outcome related salience" signal.

We are not sure if there may be a misunderstanding here. We have extensively explained in the manuscript how the polarity of SNr responses to airpuff objects which cause inhibitions similar to what is seen for good object in monkey B (Fig 2F) is consistent with the salience coding theory given the similar positive salience of airpuff and good objects in monkey B and despite their opposite valence (Fig 2D). In monkey R, where salience of airpuff is more similar to neutral objects so is firing rate to airpuff more similar to firing to the neutral objects (Fig 2E-G). Nevertheless, in both monkeys, airpuff objects despite having a lower value compared to neutral (based on choice see Fig 2D-E), has a firing rate that is less than firing to neutral which is consistent with salience and not value coding (i.e. if value coding was true we should have seen a stronger excitation to airpuff compared to neutral objects in both monkeys given its lower valence compared to neutral)

The positive correlation in the scatter plots shown in Fig 3 is just quantifying the above population average observations on individual neurons. We are reporting extensive statistics in the caption of Fig 3 supporting all the claims.

Example raster plots for the different valence conditions would help appreciating the saliency response

We have presented extensive raster plots for the value dimension in our previous publications (e.g. Ghazizadeh and Hikosaka, Science Adv 2021 Fig 2,4,S8 and S3).

To address this comment, we have now added example neurons in value, novelty and aversive dimensions in supplementary Fig S1&S7.

7) The abstract is not clear. There are multiple grammatical errors. There are concepts used that lack clarity like "saliency memory", and should be described/introduced more explicitly.

By 'saliency memory' we mean attentional bias toward an object due to past experience even in the absence of any current outcome during free viewing (i.e. object keeping memory of its past saliency in the absence of reward or punishment or their expectations)

Since abstract has word limitations, we have added our working definition of saliency memory in the introduction section page 2 line 35-38:

"Furthermore, since this attentional bias happens due to past experience and can be seen in the absence of rewarding or punishing expectations in free viewing, we refer to it as saliency memory of an object."

We have fixed a few typos in the abstract.

8) The results section lacks statistical significance values. It is written in a descriptive style with multiple assertions like "...neurons on average showed stronger excitation ..." having no p value and test specified that would support the statement or "...aversive signal and value signal were positively correlated in both vIPFC and cdLSNr ..." without providing the correlation coefficient and significance value. This issue is pervasive throughout the results section and is not ameliorated by the statistical results accompanying some figures as many statements in the text are made without explicit link to a figure panel. To gain confidence in the results and understand its reliability it will help to have statistical data for each assertion and an explicitly mentioning of which test statistics has been used for the inferential statistics.

The results reported in most cases are accompanied with extensive statistical tests reported in the figure captions rather than the main text.

For instance, for the sentence mentioned by the reviewer "...neurons on average showed stronger excitation ..." please see detailed statistics in caption of Fig 1G which reports statistics and p-values for the marginal distributions of value and novelty AUCs across all neurons.

9) Related to the previous point, the authors may want to provide a direct statistical comparison of vIPFC versus cdlSNr findings whenever the difference between areas is interpreted.

We have now added a supplemental figure (Fig S14) comparing onsets of value, novelty and aversive signals between the two regions and report statistics in the caption.

In one place where we can have compared value signal in SNr and vIPFC we have now added the statistical test in the text page 4 line 8-10:

“SNr showed robust excitation to bad objects and an equally robust inhibition to good objects with drastic differential response to good/bad objects (significantly stronger than vIPFC, $t_{233}=12$, $p<1e-3$).”

10) Figures (Fig. 1C,2E) of average firing rates are presented without error bars / error shading, which is not appropriate as it does not allow understanding how typical the average (shown) response is. Error bars should be included. Similarly, the text provides at multiple times average values (e.g. percentage difference in firing) without giving standard errors/standard deviations across the neural populations. This should be added.

We have now added errorbars representing standard error of mean (sem) to all PSTHs when we show population averages in all main and supplemental figures.

We have added \pm sems values in the page 4 lines 17-18 where we report average values.

11) The expectation was that the introduction introduces the literature on how appetitive, aversive, novelty information is encoded in vIPFC and how this differs to the basal ganglia. This literature is missing and is not provided. A rich existing literature is not mentioned.

We have now added a reference to previous literature on novelty, value and aversive coding in PFC and basal ganglia in the Introduction section page 2 line 16-20:

“Prefrontal cortex is known to be sensitive to object novelty (10, 11) even from childhood (12) . Prefrontal cortex is also implicated in processing rewarding and aversive stimuli (13–15) . Likewise selective coding of rewarding and/or punishing stimuli is observed in basal ganglia (3, 16, 17). In addition, novel objects are also known to evoke enhanced responses in some areas within basal ganglia such as in caudate(18).”

And in page 2 line 27-38:

“It is often assumed that reward, novelty or punishment enhance an object’s importance and relevance for an animal both motivationally and emotionally (aka object’s salience). However, in many cases such physiological reactions are just presumed and not directly measured and the exact meaning of salience is not fleshed out (16, 20, 21). In this work, by salience we mean attentional bias toward an object which is quantified by measuring gaze bias toward an object during free viewing of multiple competing objects (5). Attentional bias conceptualization of

saliency has been used previously (3, 22, 23). In this case, if novelty and aversiveness are found to affect the gaze bias behavior and the visual responses in neurons in the same manner, one may conclude that the neurons are involved in coding objects attentional saliency rather than their valence. Furthermore, since this attentional bias happens due to past experience and can be seen in the absence of rewarding or punishing expectations in free viewing, we refer to it as saliency memory of an object.”

12) The result section does lack specific basic information:

- p2 line 37: how many sessions in each monkey?
- p3, line 1: how many fractal objects were used in each category on average?
- p 3 line 19L how many previous trials or encounters were done for familiar objects (on average \pm SE) ? Across how many sessions on average (the methods only states >10 session per set). This information is needed to understand how familiarity and long term saliency memory is defined.

About the number of sessions for reward /aversive /familiarity training, we had at least 5 sessions for each fractal prior to start of recording (The number of trials in each session per fractal is detailed in the method section). Obviously, since these objects were used for the 2 years duration of the recording in both regions in passive viewing, and reward/aversive training the total number of sessions was much higher by the end of the study.

We have now added the exact number of object seen across each category for each monkey in method section under the ‘stimuli’ heading page 19 line 31-38:

“Monkeys saw many fractals across appetitive, aversive and perceptual domains. For the good/bad sets, monkey B and R saw 96 (12 sets) and 104 (13 sets) objects in good/bad sets (half good/half bad, Figure 1A). For novel/familiar sets, both monkeys had 8 familiar objects. Monkey B saw 360 and monkey R saw 580 novel objects for these recordings (Figure 1A). For aversive sets, each monkey B and R saw 3 sets for each airpuff, saline and timeout types (24 objects in total for each monkey (Figure 2,4). Finally, for neural responses shown in Fig S6, data was from 4 amount and 4 probability sets in monkey B (40 objects in total) and 6 amount sets and 4 probability sets for monkey R (50 objects in total). Overall, monkey B saw 528 objects and monkey R saw 766 objects across all object types reported in this study.”

minor

abstract:

- shows or has shown instead of ‘...evidence show...’
- “visual responses within neurons” ... ‘of neurons’

Both fixed

- line 20 the same or different population of neurons
Changed to ‘the same neurons’

- p. 7 "... two dimensions (value, aversiveness, novelty) ..." 2 or 3 dimensions?

We meant to say pairwise correlation between two out of three dimensions were done.

Changed to:

'between every two dimensions from among value, aversiveness, novelty were performed'

there are multiple grammatical errors that are not listed individually here. The errors continue in the methods section ...e.g. is in "T2 relaxation times in much shorter in SNr area" or singular in "a scleral search coils",...

We have tried to proofread the manuscript and fixed the above mentioned issues.

Reviewer #2 (Remarks to the Author):

This is an interesting study that sheds new light on the role of the vIPFC and the SNr in maintaining the memory of value novelty and aversiveness. However, in my opinion, the Authors should show additional evidence in order to support their hypothesis.

Major Points
The description of the training tasks, which is clear in their previous paper (Ghazizadeh A., et al, 2021), here is confusing and difficult to understand. The Authors should rephrase the description of the tasks in the Method section.

We have made a few modifications to the description of the training tasks in the method section (highlighted). As we have used similar descriptions in our past publications it was not immediately clear what portions the reviewer found ambiguous. If our current clarifications are not sufficient, we appreciate more specific comments so we can better clarify.

The interpretation of the passive viewing tasks relies on the behavior in the training tasks. However, analysis of the behaviors is not sufficient. For instance, the report only the results of the choice made by the monkeys in choice and in the free viewing tasks. As discussed below, the analysis of the saccade latency is a good measure to establish the behavioral relevance of an object.

We thank the reviewer for the comment. We have now added the saccade latency during training of aversive sets in supplementary Fig S4. As expected, saccade latency to good was much faster than neutral. For the aversive objects saccade latency to airpuff was also fast similar to good objects but saline and timeout objects had slower saccade similar to neutral objects. The fast saccade latency toward airpuff objects was consistent with their positive salience. These findings are now described in the figure caption and the result section page 5-6:

“Analysis of saccade reaction time for airpuff, neutral and good objects also showed faster saccades toward airpuff objects compared to neutral objects similar to saccade reaction time to good objects consistent with attentional salience observed in free viewing (Fig S4).”

and in page 7 line 19-21:

“Consistently, the saccade reaction times to saline and time-out object was not different from neutral object and showed even a trend to be slower than neutral objects (Fig S4).”

In a previous study (Ghazizadeh A., et al, 2021) the authors found a strong memory value and an amount and probability reward memory signals in both areas. They also found a stronger activity for objects with uncertain reward probability compared to objects with 0

and high reward probability in vIPFC and a gradually stronger inhibition with the increasing reward amount and expectation and a slight excitation for small reward amount and for 0 reward probability in SNr (Figure 5 A- D of Ghazizadeh A., et al, 2021). Here there are 4 types of objects associated with 0 rewards. The novel and the familiar objects (the task of figure 1) and the neutral objects (figure 2). In both the familiar and the neutral objects trials, the probability of reward is zero, while in novel object trials, there may be uncertainty about the reward. Isn't possible that the familiar and novel objects are instead associated with expected reward?

The neurophysiological data in this study may support the hypothesis. In this study, the neural response of vIPFC neurons to the familiar objects is lower than the response to novel objects, as the response to 0 reward probability vs the response to the uncertain reward in their previous study (Ghazizadeh A., et al, 2021). Likewise, neurons in SNr respond with an increase of activity to familiar and novel objects, as in the low amount and low reward probability. The same observations can be drawn from Figure 1E of both vIPFC and SNr. In vIPFC, the higher the activity of the Bp neurons, the higher the activity for the familiar objects. In other words, the results might be explained by the gradual increase of the response with the expected reward amount and probability: familiar (no reward), uncertainty (novel stimulus), low reward, and good reward. The authors should clarify why the familiar and novel objects do not have any expected (low or uncertain) reward value and report the latencies of the saccade to good, bad, familiar, and novel in the choice tasks.

This is indeed an important observation by the reviewer. In real life novel object often carry outcome uncertainty with them. However, it is not immediately clear why this uncertainty should only be related reward expectation as in our experiments as well as in real life we encounter novel objects that end up being aversive.

In addition, we also would like to emphasize that while the correlated activity for value and novel objects in vIPFC may leave the interpretations open for value uncertainty in novel objects, the lack of a strong population response in SNr to novel objects violates value related interpretation for novel/ familiar objects especially given the fact that normally responses in SNr are observed to be stronger than vIPFC when value memory is involved. Furthermore, in SNr a value related interpretation predicts a higher excitation to familiar objects (which has zero reward) compared to bad objects (which had small reward) however the opposite is observed (Fig 1D).

Finally, we want to emphasize that in our paradigm the monkeys always saw the novel objects in the novel/familiar passive viewing context and these novel objects would never be used in the future in any other task or condition. Our monkeys were very well trained in a such a context for many months and many sessions prior to start of electrophysiology.

Therefore, while we cannot completely rule out reward/aversive expectation toward novel/familiar objects, based on the above arguments we think such an effect is unlikely to affect our results. Nevertheless, we are now explicitly mentioning this possibility and our arguments in the discussion section page 10 line 25-35:

“In real life, novel objects often carry outcome uncertainty with them. Given our previous report on coding of uncertainty value memory in vIPFC and SNr, one interpretation of novelty responses seen in vIPFC can be that those novel objects signaled a possibility of reward. However, the lack of a strong population response in SNr to novel objects violates such value-related interpretations for novel objects especially given the strength of SNr responses when value memory is involved. Furthermore, in SNr a value-related interpretation predicts a higher excitation to familiar objects (which has zero reward) compared to bad objects (which had small reward) which is the opposite what is observed (Fig 1D). Given these observations and the fact that in our paradigm novel objects seen in novel/familiar context were never used before or in the future, it is unlikely that the novelty responses observed can be reduced to reward or outcome uncertainty.”

Figure 1D shows that in SNr, the onset of the novelty signal is significantly later than the onset of the value signal. However, in the supplementary figure (S2 that is S1) it seems that only in monkey B the onset time of novelty signals was later than the value signal. If the onset of the novelty is later than the value only in monkey B, the authors should report it.

Indeed, both monkeys showed this effect clearly (i.e. value and novelty onset not different in vIPFC but novelty onset being later than value onset in SNr). We have now added the onset CDFs separately for each monkey in new panels Fig S2.

I found the section regarding the results of aversive signals muddled and difficult to read. From what I understood, the choice task is a decision-making task, in which the monkeys can choose only one of two objects, while in free viewing there is no such constrain because monkeys are not required to make any choice.

This is correct. “the choice task is a decision-making task, in which the monkeys can choose only one of two objects, while in free viewing there is no such constrain because monkeys are not required to make any choice” Furthermore, in free viewing there was no outcome for looking at fractals.

The authors should report the latencies of the first saccades in free viewing and choice tasks in a supplementary figure and compare the latencies of the first saccades to good objects with the latency of the first saccade to aversive and neutral objects. If there is not a significant difference, I would expect that in the choice task a significantly longer latency of saccades to good objects when paired with aversive than when paired with neutral objects since in the choice trials there is a strong competition between the two objects.

We have now added saccade reaction times to aversive object in Pavlovian task force trials (Fig S4). There we see significant effects consistent with positive salience in airpuff objects and lack of salience for saline and time-out objects. Consistent with this result, we are showing significant RT facilitations to good and novel objects compared to bad

and familiar object in a free looking task which we are showing as in Fig R2 in this document for reviewer 1.

We did not observe clear RT differences in free viewing for any of our set types. We have analyzed free viewing results across dimensions in detail in Ghazizadeh et al Frontiers, 2016. Many metrics such as first saccade, object scanning and view duration is significant in this task but in general RT is not strongly affected.

Also as per the reviewers suggestion, we looked at the reaction time in choice trials in the aversive set, but do not see significant differences among conditions. The only trend we observe is a slower choice when aversive and neutral objects were paired in all three set types (airpuff, saline, time-out).

The authors should plot the difference between the response to good objects and the response to aversive objects in the value vs aversive signals plot (Fig 1E) to better compare the behavior with the neural activity in vIPFC and SNr, separately for the two monkeys.

We assume the reviewer was referring to Fig 2F-G (Fig 2E in the last version of the paper). Here the value and aversive (airpuff) signals were already shown separately for each monkey in Fig 2F-G and the value vs aversive plots in Fig 3 are also separately shown for each monkey.

For results of aversive (saline and time-out) shown in Fig 4-5 which is collapsed across monkey, we are showing each monkey separately in Fig S5.

If the reviewer had some other figures in mind, we would appreciate a clarification.

Minor points

In this study, the visual response was evaluated in a time window from 200 to 600 ms after stimulus onset, while in the previous study it was calculated from 100 to 400 ms after the object. Is there any particular reason why they choose two different epoch windows?

Thanks for the comment. As we show below our main results are robust to the choice of time window.

The reason for using 200-600ms window in Fig 3 and 5 is the very strong value signal in SNr to the GN (good minus neutral) in the aversive sets. As mentioned in the methods we had to use 1.5 times larger reward for good objects in the aversive set compared to good/bad set to guarantee monkey collaboration in aversive sets. In addition, neutral objects had no reward compared to bad object small reward thus good minus neutral had a much stronger value signal compared to good minus bad which was analyzed in the previous paper. Since maximum of absolute AUC is equal to 'one' using the 100-400 ms saturated the AUC for many SNr neurons such that correlations for TN, SN and PN with GN was not meaningful. We are showing an example of scatter plot using 100-400ms vs

200-600ms for comparison. (Note the saturation of SNr GN signal on the x-axis when using 100-400ms window)

Figure R3.a

Therefore, we decided to use a rather shifted and wider window to allow for more accurate calculation of aversive vs value correlations in SNr. To be consistent we used the same 200-600ms in all of our analysis shown in Fig 1, 3 and 5. Nevertheless our results are robust to the choice of time window (we are showing the novelty-value correlations in 100-400 below for comparison with the paper)

Figure R3.b

Please note that the results shown for pairwise correlation between value, novelty and aversiveness in Fig 6 are done with the 100-400ms window with findings that are consistent with Fig 1,3,5 which used 200-600ms window. The reason that we could use the 100-400ms in Fig 6 is because here we correlated the value signal from the GB sets which do not saturate SNr AUC with novelty and aversive signals in novel and aversive sets (compared to GN signal from aversive sets which were not useful for correlation as explained above)

Fig S5: the range of the blinking rate should be the same (0 to 0.6) for both monkeys
The current range is chosen to better show the differences among set and object types within each monkey rather than comparison between monkeys.

There are two supplementary figures 2.

This is corrected.

The way the subplots in each figure are named is confusing and difficult to follow. For instance, the subplots 1C of figure 1, should be broken into two: figure 1 C (vIPFC) and 1 D (SNr). The same for figure 1 D and 1 E, and for figures 2 D and 2 E.

The panels are updated.

Reviewer #3 (Remarks to the Author):

Here, the authors present a paper that extends their work on object-based memories stored in cortico-basal ganglia circuits to include objects that are not reward associated. I think the authors' intents here are noble, and I believe the experiments will lead to important insights, but in its current form I have several major concerns about the conceptualization and implementation of the research described in this paper. The major concerns revolve around the conceptualization of salience and value, how we measure these concepts, and on the amount of weight that is attributed to the results from monkey R. To the authors, the neural data supports 'salience' coding, rather than value coding. However, the lack of clear definitions, the crude measurement of value, and the correspondence between the behavioral and neural data from monkey R regarding the airpuff raises serious challenges to their interpretation. Accordingly, this is valuable data to show but the interpretations should be strongly moderated. In detail:

We thank the reviewer for the important comments. Below we aimed to address each point.

1) A clear definition of 'salience' is lacking from this paper. Elsewhere, I have read about visual salience, attentional salience, physical salience, motivational salience, emotional salience, psychological salience; the list goes on and on. These phrases do not all mean the same thing. For example, a loud bang caused by a passing car might be physically and attentionally salient, to a pedestrian, but it is not likely to be motivationally salient (Or maybe it is! The point is these concepts are separable). This lab drives the salience narrative in cortico-basal ganglia neurophysiology, and yet it is unclear what they mean by 'salience'. In figures 1-3, I inferred they are talking about motivational salience. Then, in figures 4 and 5, and in the associated text, they switch from using airpuff as punishment to using bitter tastes and time-outs. They write, on Line 18, page 6, '[t]o further test this hypothesis, object punishment history was created by using aversive outcomes (aversive taste or time-out) that did not change object salience (Fig 4A).' This suggests that they do not mean motivational salience, but rather physical or attentional salience. (To reinforce that interpretation, they go on to show that the animals avoid cues paired with aversive tastes and time-outs, a behavior consistent with increased motivational salience for cues associated with those objects). After reading figures 4 and 6, I looked back at figures 1-3 and try to re-interpret them regarding physical or attentional salience, but it doesn't fit. I am left with the impression, frankly, that the amorphous definition of salience is required to make the 'salience theory' work. It would be much better if in the introduction the authors created a working definition of salience and apply it consistently throughout the manuscript.

We thank the reviewer for the important comment. It is true that term 'salience' in the field is used to describe different phenomena. However, in our case whenever we say salience we mean 'attentional salience' which is always corroborated by free viewing gaze bias consistently in all dimensions including value, novelty and aversiveness. Thus, in our working definition object A is more salient than object B if there is gaze bias toward object A compared to B in the absence of any instructions (when there is no instruction

to look at or avoid the objects and no outcome delivery which is true in free viewing). Based on this definition we declared saline and time-out objects as having slightly negative and airpuff as having positive salience when compared to neutral based on results presented in Figs 2, 4, S6. Same is true in comparison of novel vs familiar and good vs bad (Fig S8)

We have now added a paragraph acknowledging the existence of various definitions for salience in use in the literature and explicitly declaring our working definition in Introduction section page 2 lines 27-38:

“It is often assumed that reward, novelty or punishment enhance an object’s importance and relevance for an animal both motivationally and emotionally (aka object’s salience). However, in many cases such physiological reactions are presumed without direct measurements and the exact meaning of salience is not fleshed out (16, 20, 21). Here, we consider a precise working definition of salience as the attentional bias toward an object which is quantified by measuring gaze bias during free viewing of multiple competing objects (5). Such attentional bias conceptualization of salience has been used previously (3, 22, 23). In this case, if novelty and aversiveness are found to affect the gaze bias behavior similar to value one may conclude them to have a positive salience. Furthermore, since this attentional bias happens due to past experience and seen in the absence of rewarding or punishing expectations in free viewing, we refer to it as salience memory of an object. In such a case a neuron is encoding salience if it responds similarly to objects with similar salience regardless of their valence.”

2) Choice and value are related but not equal. Choices (here) are binary, whereas value is not. Choices are equal to value passed through a softmax function, or some other kind of thresholding operation. This poses a major challenge to scientists that want to measure value. Here, the authors argue that the neural responses which they recorded reflect salience rather than value, largely because they look more like the free viewing behavior than the choices. However, if we agree that choice and value are not equal, and we should because it is obvious, then the clear interpretation of this data as coding salience rather than value is hard to substantiate.

The reviewer raises an important yet subtle point. It is true that choices are binary and that mapping from value to choice often goes through nonlinear transformations of the utility function and then a softmax like function to predict choice percentages (subjects of study for neuro-economists). However, such nonlinearities do not affect our main argument which is only concerned with the ORDINAL value of two objects (i.e. whether object A is more or less valuable than B and not the magnitude of their value difference). That is as long as our agents are rational such that in cases where value of A is larger than B, they choose object A over object B and if ordinal value $C > B > A$ then choice percentage of C over A is higher than choice percentage of B over A (transitivity of preference), we can compare this ordinal value which we get from choice with the rank order of the firing rate response to those objects. As we explain below these ordinal

comparisons show that the attentional salience rather than value is consistent with the observed firing patterns.

Furthermore, the differences between monkeys B and R regarding the airpuff are interesting, and I believe they speak to this point. The lower left-hand panel for the cdlStr responses from B and R, in Figure 2E, perfectly correspond to the behavioral differences between the two monkeys. These neural differences are entirely consistent with a subjective value signal. The marginal distributions shown in Figure 3 and 5 are also consistent with subjective value, the only difference is a positive code in the PFC, and an inverted code in SNr.

As for the figure in reference by the reviewer (Fig 2): the firing rates in both monkeys B and R, in old Figure 2E (now Fig 2F-G), perfectly correspond to the behavioral differences between the two monkeys if one is to look at their gaze bias in free viewing BUT NOT their choice (Fig 2D-E). In both monkeys the choice between airpuff and neutral is biased toward neutral object (more in monkey B and less in monkey R but nevertheless >50%). If vIPFC neurons were coding value then the firing to airpuff should be smaller than neutral but in both monkeys the airpuff firing is 'larger' than neutral. Conversely, If SNr neurons were coding value then the firing to airpuff should have been larger than neutral but in both monkeys the airpuff firing is 'smaller' than neutral.

On the other hand, the order of firing in both monkeys is more consistent with their gaze bias. In monkey B gaze bias to airpuff is very positive so is the puff minus neutral signal is more positive in vIPFC and more negative in SNr (polarity in SNr is always the opposite of vIPFC). In monkey R gaze bias to airpuff is only slightly positive so is the puff minus neutral signal is slightly positive in vIPFC and slightly negative in SNr.

These points are already discussed when talking about Fig 2 in the Results section However to clarify the important point about difference between value and choice raised by the reviewer, we have added a cautionary note in page 6 lines 20-24:

"Note that our arguments about valence only rests on deciphering the 'ordinal' value of objects based on binary choices (i.e. if object A is chosen more than object B, object A has more subjective value compared to B) and does not require knowledge of their magnitude of their actual utility or subject value differences which cannot be estimated from binary choices without additional assumptions or experiments."

3) The authors claim that the 'salience theory' explains 'peculiarities in neural responses' when risk is involved (page 6, line 35 and Fig. S6), but they do not provide enough details about those experiments to back up this claim. If the monkeys are risk seeking – and nearly all monkeys are risk-seeking for small rewards – then subjective values and risk attitudes explain S6 as well as the 'salience theory'. Here, as elsewhere in this manuscript, the population PFC codes value in a positive way (good higher than bad) and

the population cdlStr codes value in an inverted fashion (bad higher than good). Furthermore, displaying these signals as a difference, as in Figure S6c, makes these neural responses look like the uncertainty responses the senior author showed in the septum (Monosov and Hikosaka, NatNeuro 2013 – one of my favorite papers of all time!). However, they are not pure uncertainty signals in the way the septum signals are. These responses are consistent with subjective value, not pure uncertainty.

We apologize for being a bit cryptic here. The reviewer is correct that stand alone the results of amount and probability set can also be explained by subjective value of risk. This was indeed close to our conclusion in our last paper (Ghazizadeh et al Sci Adv 2021, Fig 5, Fig S6-7) where we first reported these results (but please see discussion of some effects such as range expansion in firing and deviation of both convex utility and prospect theory in that paper which adds some complexity to risk interpretation).

What we meant to say here is that given the new observations about responses to valueless novel and aversive airpuff objects, the previously reported responses to probability and amount sets could be also consistent with our new suggestion about coding of attentional salience in vIPFC and SNr. The specific peculiarity we referred to was that the bell-shaped curve which results from the subtraction of firing response to amount from probability shows a slightly negativity in vIPFC for the lowest and highest values (i.e. 0% and 100% objects in amount set being lower than probability set despite matching value and no risk) and the opposite in SNr. This effect can also be seen if attentional bias to amount and value are subtracted in free viewing but has no easy interpretation in predictions made within value or risk frameworks.

We have now rephrased this paragraph and clarified the peculiarity we referred to in the page 7 lines 30-35:

“The salience theory is also consistent with neural responses in vIPFC and cdlSNr to objects with graded reward amount and probability reported previously (19). In both cases, neural firings paralleled objects learned salience including the enhanced attention to uncertain rewards and lower attention to amount vs probability objects at the lowest and highest value extremes despite matching value and lack of uncertainty as measured by free viewing (Fig S9).”

We have also added arrows to Fig S9 to annotate the negativity discussed above and added further clarifications in the figure caption.

4) Page 5, Line 35 is very confusing. I can't understand what they meant to write, so I will just leave this for the authors to address.

We believe the reviewer is referring to the paragraph starting with “These observations make interesting predictions ...”

What we say here is exactly the related to the reviewer's comment 2 above. The observed difference in firing rate puff minus neutral is consistent with variation of salience (gaze bias in free viewing) and not choice. To be consistent with choice where both monkeys show value of neutral > value of airpuff, we should have seen higher firing to neutral than airpuff in vIPFC and lower firing to neutral than airpuff in cdISNr which we do not see. We have clarified this point in page 5 lines 19-23 as following:

“Thus, in neither monkey, we observed responses that were consistent with the negative valence of the airpuff objects. To be consistent with value since both monkeys showed negative value for airpuff compared to neutral (value of neutral > airpuff from binary choice, Fig 2D-E), we should have seen higher firing to neutral than airpuff in vIPFC and lower firing to neutral than airpuff in cdISNr but the opposite is observed (Fig F-G).”

5) The circuit cartoon in figure 7 does not include a role for dopamine responses in novelty processing, and this omission is not consistent with many previous studies that have shown very strong novelty responses in dopamine neurons, and a role for those novelty responses in preventing latent inhibition. Here, if I read figure S2 correctly, the authors recorded 9 dopamine neurons. Can the authors explain why they see less dopamine novelty responding than previous studies? Is there a reason the field should re-assess the widespread understanding that novelty drives dopamine responding?

We thank the reviewer for raising this very important point. Despite the widespread assumption about the role of DA in novelty processing to-date we are not aware of any well-controlled study that shows “pure object novelty” response in dopamine neurons. We previously had now a short discussion of this finding in the discussion which we are now expanding with additional references in page 9 lines 29 onward:

“Lack of novelty coding in putative DA neurons may seem rather curious. But despite the widespread belief about the role of DA in novelty processing, most previous studies on DA-ergic activity could not fully dissociate stimulus novelty from confounds such as recency/sensory surprise (31), reward expectation and learning (32, 33) and movements such as orienting (34) and sniffing (35). Furthermore, studies that manipulate novelty seeking with DA depletion, agonist and antagonists (36) in addition to being often confounded with concurrent changes in locomotion may not be relevant here since DA firing does not always correlate with DA release in the target area(37). Other studies with controls for these confounds are based on fMRI signals in SNc which do not afford the resolution to infer DA neural firings (38). Interestingly, a recent study with a well-designed design confirms this lack of object novelty signal in DA neurons as well as the lateral habenula as one of the main inputs to DA neurons (39).”

Minor

1) In previous papers, this lab has used 40psi stimuli, but there is no indication of airpuff parameters used here, that I can see.

Here we have used 100 ms of 8-psi airpuff delivered to the right eye. This is now added to the method section page 21 line 25.

2) Page 2 line 29 'paralleled measured' seems like a typo. There are many other places where the writing could be improved.

Fixed to "paralleled the measured".

We have read the whole paper and tried to catch other typos hopefully copyeditors can help if there are any remaining ones.

REVIEWERS' COMMENTS

Reviewer #1 (Remarks to the Author):

the authors made impressive revisions and clarified all aspects I raised. This manuscript has the potential to be highly influential.

Reviewer #2 (Remarks to the Author):

The Authors have addressed all of my concerns with the original manuscript. The revised manuscript is ready for publication

Reviewer #3 (Remarks to the Author):

1. The authors adequately address my concern about salience definition

2. I agree with the authors that the brain signals are not coding 'ordinal' value.

3. The authors claim that their lack of dopamine novelty signaling makes sense, and in support they cite a recent paper that purports to show 'value-less' novel cues do not activate dopamine neurons (T. Ogasawara, et al Nat. Neurosci. 2022). This citation is one paper – set against many, many unacknowledged papers from top labs that show clear novelty responses (PMIDs 32027824, 27787196) – and I do not see how it is relevant here. Unlike in the Ogasawara paper, the current data does not show that the novel cues were 'value-less.'

I have no further comments or concerns

Reviewer #3 (Remarks to the Author):

3. The authors claim that their lack of dopamine novelty signaling makes sense, and in support they cite a recent paper that purports to show ‘value-less’ novel cues do not activate dopamine neurons (T. Ogasawara, et al Nat. Neurosci. 2022). This citation is one paper – set against many, many unacknowledged papers from top labs that show clear novelty responses (PMIDs 32027824, 27787196) – and I do not see how it is relevant here. Unlike in the Ogasawara paper, the current data does not show that the novel cues were ‘value-less.’

We agree with the reviewer about the controversial nature of our claim about DA neurons not encoding stimulus novelty. We believe that if true this point is very interesting and deserves investigation hopefully by other independent groups in the future. To address the reviewers concern we have added a qualifier to tone down our claim and emphasize need for further investigation in page 10 line 2 as following:

“Additional studies will be required further verify lack of novelty coding in DA neurons.”

We also would like to point out that we have already cited Morrens et al 2020 Neuron (PMID 32027824) . But as we said in the discussion we believe it is very hard that the authors can rule out the effect of sniffing on the DA responses despite their effort to show lack of partial correlation in Fig S1 (for example presence of nonlinear effects could not be ruled out by their method). Furthermore, the PMID 27787196 (Lak et al 2016) only shows recency coding and does not address novelty.